# The spatiotemporal characteristics of the air pollutants in China from 2015 to 2019

**Peng Guo** [*], **Aminat Batalbievna Umarova, Yunqi Luan**

Department of Soil, Moscow State University, Moscow, Russian Federation

* 956095891@qq.com

**Data Availability Statement:** All relevant data are within the manuscript and its Supporting Information files.

**Funding:** Author(s) received no specific funding for this work.

## Abstract

China's rapid industrialization and urbanization have led to poor air quality, and air pollution has caused great concern among the Chinese public. Most analyses of air pollution trends in China are based on model simulations or satellite data. Studies using field observation data and focusing on the latest data from environmental monitoring stations covering the whole country to assess the latest trends of different pollutants in different regions are relatively rare. The State Council of China promulgated the toughest-ever Air Pollution Prevention and Control Action Plan (Action Plan) in 2013. This led to a major improvement in air quality. We use the hourly Air Quality Index (AQI) and mass concentrations of $PM_{2.5}$, $PM_{10}$, CO, $NO_2$, $O_3$, and $SO_2$ in 362 cities from 2015 to 2019, obtained from the Ministry of Ecology and Environment, to study their temporal and spatial changes and assess the effectiveness of the policy on the atmospheric environment since its promulgation and implementation. We found that the national and regional air quality in China continues to improve, with $PM_{2.5}$, $PM_{10}$, AQI, CO, and $SO_2$ exhibiting negative trends. However, $O_3$ and $NO_2$ pollution is an urgent problem that needs to be solved and the current control strategy for $PM_{2.5}$ will only partially reduce the $PM_{2.5}$ pollution in the western region. Although the implementation of "Action Plan" measures has effectively improved air quality, China's air pollution is still serious and far from the WHO standard. Implementing measures for continuous and effective emissions control is still a top priority.

## Introduction

To reduce pollution, considerable effort and resources must be invested to implement the measures in place. The evaluation of the effectiveness of the promulgated measures can provide important information for China and other developing countries and highly polluting countries to formulate high-efficiency air quality policies. Recent studies often use model simulation methods to study pollutant changes, for example, Zhang et al. quantified the emission reduction of air pollutants by examining each measure of the "Action Plan" [1] using models. The authors estimated that the national population-weighted concentration of annual mean $PM_{2.5}$ decreased from 61.8 μg/m³ to 42.0 μg/m³ from 2013 to 2017 [2]. Cai et al. used the model to simulate the concentration of $PM_{2.5}$ in the Beijing-Tianjin-Hebei area from 2012 to 2020 and concluded that the concentrations of $PM_{2.5}$ in 2017 and 2020 will be 28.3% and

**Competing interests:** The authors have declared that no competing interests exist.

37.8% lower, respectively, than in 2012 [3]. Jiang et al. used model simulation to conclude that, from 2012 to 2017 in the Pearl River Delta, the "Action Plan" effectively reduced $SO_2$ by 34%, NOx by 28%, $PM_{2.5}$ by 26%, and volatile organic compounds by 10% [4]. Zheng et al. [5] and Xue et al. [6] used similar methods to obtain similar conclusions.

In addition, many previous studies have used satellite retrieved aerosol optical depth to estimate trends in $PM_{2.5}$ concentrations. Peng et al. reported that China's $PM_{2.5}$ concentration increased significantly between 1999 and 2011, especially in central and eastern regions of China. The proportion of areas with $PM_{2.5}$ concentrations higher than 35 µg/m$^3$ increased year by year, and the areas with $PM_{2.5}$ concentrations lower than the annual primary standard of 15 µg/m$^3$ decreased continuously [7]. Ma et al.'s spatiotemporal trends of $PM_{2.5}$ based on satellite studies show that the annual mean between 2004 and 2007 increased by 1.97 µg/m$^3$, while the annual mean between 2008 and 2013 decreased by 0.46 µg/m$^3$ [8]. Krotkov et al. observed through the Ozone Monitoring Instrument (OMI) of NASA's Aura satellite that the North China Plain has the most severe sulfur dioxide pollution in the world, but it has been on a downward trend since 2011. Due to economic slowdown and government intervention, efforts to curb emissions from the power and industrial sectors led to reductions of approximately 50% from 2012–2014 [9]. Verstraeten et al. used satellite observations to show that from 2005 to 2010, China's ozone ($O_3$) concentration has been increasing steadily at a rate of 7% per year [10].

There are also many reports on the spatial and temporal changes in gas and particulate pollutants in China, but most of them are limited to a city or a certain type of pollutants [11–16]. For example, Zhou et al. studied the spatiotemporal changes in air quality in Beijing [11], Guo et al. used ground observations to assess the $PM_{2.5}$ concentration and exposure throughout China [12], and Chen et al. reported spatiotemporal changes in $PM_{2.5}$ and their relationship to meteorological factors in Nanjing City [13], among many other studies.

Some studies have used monitoring networks to study the spatiotemporal changes in multiple pollutants at multiple sites (urban). For example and Silver et al. reported that substantial changes in the air pollutants $PM_{2.5}$, $SO_2$, $NO_2$ and $O_3$ occurred across China from 2015–2017 according to data from 1689 sites [17]. Guo et al. studied the air quality of 366 cities in 2015–2017 and the spatiotemporal changes in six air pollutants, i.e., $PM_{2.5}$, $PM_{10}$, $SO_2$, CO, $NO_2$, and $O_3$, and AQI [18]. The most recent trends in pollutant changes have also exhibited changes. For example, both the present study and that conducted by Guo et al. found that $NO_2$ hardly changed from 2015 to 2017.

Here, we use the latest data from environmental monitoring stations covering the whole country to evaluate the most recent trends of various pollutants in different regions of the country.

## Materials and methods

The hourly data on national pollutants were downloaded from http://beijingair.sinaapp.com/, which were obtained from http://pm25.in. The URL http://pm25.in provides information identical to that provided by the official Ministry of Ecology and Environment (http://106.37.208.233:20035/). Other studies have used similar data (for example, Silver et al. 2018 [17], Guo et al. 2019 [18], Fan et al. 2020 [19], Rohde and Muller 2015 [20], Liang et al. 2016 [21], Leung et al. 2018 [22]).

We conducted a statistical analysis of the $PM_{2.5}$, $PM_{10}$, $SO_2$, CO, $NO_2$, $O_3$ and AQI in different regions every year from 2015 to 2019.

The total number of monitoring points in different regions every year N is denoted by n. Let $x_i$ be the $i$ th sample and $w_i$ be the $i$ th weight; the annual mean of the pollutants in different

regions every year is calculated using the following formula:

$$\bar{x} = \frac{1}{\omega} \sum_{i=1}^{n} x_i \omega_i.$$

If there is no weight variable, the formula is reduces as follows:

$$\bar{x} = \frac{1}{n} \sum_{i=1}^{n} x_i.$$

The standard deviation of pollutants in different regions every year is calculated as follows:

$$s = \sqrt{\sum_{i=1}^{n} \omega_i (x_i - \bar{x})^2 / d}$$

where $d = n - 1$.

The lower limit of the 95% confidence interval of the mean of the pollutants in different regions every year is calculated as follows:

$$\bar{x} - t_{(1-\alpha/2)} \frac{s}{\sqrt{n}}$$

where $t_{(1-\alpha/2)}$ is the $(1 - \alpha/2)$ critical value of the Student's t-statistic with n-1 degrees of freedom.

The upper limit of the 95% confidence interval of the mean of every region every year is calculated as follows:

$$\bar{x} + t_{(1-\alpha/2)} \frac{s}{\sqrt{n}}$$

where $t_{(1-\alpha/2)}$ is the $(1 - \alpha/2)$ critical value of the Student's t-statistic with n-1 degrees of freedom.

We also counted the minimum $x_{(1)}$ and maximum $x_{(n)}$ values, first (25%) quantile (Q1), median or second (50%) quantile, third quartile (Q3) (75%), and custom percentiles (90%, 95%, and 99%) of pollutants in different regions every year.

We counted the annual relative change of pollutants of every region as follows:

$$dx = \frac{x_i - x_j}{x_j}.$$

where $x_i$ and $x_j$ represent the absolute value at time $i$ and $j$, respectively; and $dx$ represents the relative change.

The annual average relative change of pollutants of every region is calculated as follows:

$$d\bar{x} = \frac{x_{t+1} - x_t + x_{t+2} - x_{t+1} + \ldots x_{t+n} - x_{t+n-1}}{n - 1}$$

where $x_t \ldots x_{t+n}$ represents the absolute value at time t...t+n and $d\bar{x}$ represents the annual average relative change.

We also counted the number (percentage) of cities with positive and negative annual relative changes and the number (percentage) of cities in different intervals of annual relative change.

## Results and discussion

### Spatial distribution

**$PM_{2.5}$ and $PM_{10}$.** From 2015 to 2019, the national annual mean and annual average median of $PM_{2.5}$ were 42.7 μg/m³ and 41.0 μg/m³, respectively. We divide China into 11 regions. The Northern region includes Beijing, Tianjin, Hebei, Shandong, Shanxi, and Henan; the Eastern (Yangtze River Delta) region includes Jiangsu, Zhejiang, and Fujian; the Central region includes Anhui, Hubei, Hunan, and Jiangxi; the Southern region (Pearl River Delta) includes Guangdong, Guangxi, and Hainan; the Weihe River Basin includes Shan Xi, Gansu, and Ningxia; other regions include Xinjiang, Sichuan Basin, Inner Mongolia, Qinghai-Tibet, Yungui (Yunnan, Guizhou) and northeastern China (Heilongjiang, Jilin, and Liaoning).

From a regional perspective, the Northern (58.2 μg/m³), Xinjiang (52.5 μg/m³), and Central (46.6 μg/m³) regions have the highest annual mean $PM_{2.5}$ from 2015 to 2019 (sorted by annual mean). The mean $PM_{2.5}$ in the Northern region was 36% higher than the national mean $PM_{2.5}$. Baoding (80.7 μg/m³), Xingtai (79.1 μg/m³), Laiwu (79.0 μg/m³), and Anyang (78.5 μg/m³) in the Northern region are the cities with highest annual mean (Table 1).

The Weihe River Basin (41.1 μg/m³), Eastern region (40.2 μg/m³), Sichuan Basin (40.1 μg/m³), and Northeast region (38.4 μg/m³) have slightly lower mean $PM_{2.5}$ concentrations than the national annual mean (42.7 μg/m³); Inner Mongolia (31.8 μg/m³), Southern (31.4 μg/m³), Qinghai-Tibet (29.6 μg/m³) have lower mean concentrations than the national mean; and Yungui has the lowest annual mean (25.9 μg/m³), which is 39% lower than the national annual mean. Diqing (13.2 μg/m³), Lijiang (14.5 μg/m³), and Dali (21.0 μg/m³) of Yungui, Ali (14.4 μg/m³), Yushu (16.0 μg/m³), and Lhasa (19.4 μg/m³) of Qinghai-Tibet, and Sanya (14.6 μg/m³), Haikou (18.6 μg/m³), Shanwei (24.2 μg/m³) of the Southern region are the cities with lowest annual mean.

From 2015 to 2019, the national annual mean $PM_{10}$ was 78.6 μg/m³. Xinjiang (136.8 μg/m³), the Northern region (107.2 μg/m³), and the Weihe River Basin (92.0 μg/m³) were the regions with the highest annual mean concentration from 2015 to 2019. The value in Xinjiang was 74% higher than the national relative change. Hetian (341.8 μg/m³) and Kashi (310.2 μg/m³) of Xinjiang were the cities with highest annual mean concentration.

The Central region (75.6 μg/m³), Inner Mongolia (75.0 μg/m³), Qinghai-Tibet (68.5 μg/m³), the Northeast region (67.3 μg/m³), and the Sichuan Basin (66.0 μg/m³) had lower values than the national annual mean (78.6 μg/m³); Yungui (46.2 μg/m³) and the Southern region (50.4 μg/m³) had the lowest annual mean values.

Our results showing that the highest annual mean $PM_{2.5}$ concentration occurred in the Northern region are consistent with the literature. For example, Zhang et al. simulated the concentration of annual mean $PM_{2.5}$ in China from 2013 to 2017. The findings also show that the areas with the highest $PM_{2.5}$ concentrations in China are distributed in the Northern region (Beijing, Tianjin and Hebei, and the surrounding areas), and the emission intensity in this region is the highest of any region in China [2]. Silver et al. found that the highest annual mean $PM_{2.5}$ concentration was in Hebei, Henan and Shandong Province in the Northern region from 2015 to 2017, the median concentration in both provinces was > 60 μg/m³ (we obtained an annual average median of 61 μg/m³ in the Northern region), the monitoring site of Guangdong in the Southern region (Pearl River Delta) had a low $PM_{2.5}$ concentration, while the lowest $PM_{2.5}$ concentration (20–25 μg/m³) was found in Hong Kong in the Pearl River Delta region and Tibet [17].

Our results showing that higher annual mean $PM_{10}$ concentrations occurred in the Xinjiang Province are consistent with the literature. For example, Guo et al. also found that the maximum annual mean values of both $PM_{2.5}$ and $PM_{10}$ occurred in Xinjiang Province, which

was determined to be primarily the result of mineral dust from the Taklimakan Desert [18]. Fan et al. also found that air pollution was most serious in Xinjiang Province from 2014 to 2018 [19]. However, the distribution obtained in this study was somewhat different from Zhang et al. [2]. This may be attributed to model uncertainty or the limited number of monitoring stations available in Xinjiang in this study.

Among key provincial capitals and well-known tourist cities, only one city, Sanya, has an annual mean $PM_{2.5}$ of 14.6 μg/m$^3$ that meets standard I (15 μg/m$^3$) of China's ambient air quality standard (CAAQS). Only two cities, Sanya (28.4 μg/m$^3$) and Haikou (35.0 μg/m$^3$), have annual mean $PM_{10}$ values that meet standard I (40 μg/m$^3$) of the CAAQS. Beijing ($PM_{2.5}$, 59.8 μg/m$^3$; $PM_{10}$, 88.8 μg/m$^3$), Shanghai (41.3 μg/m$^3$; 59.0 μg/m$^3$), Guangzhou (33.5 μg/m$^3$; 54.3 μg/m$^3$), and Shenzhen (26.2 μg/m$^3$; 44.4 μg/m$^3$) all exceeded the first-level standard. During the 2014 analysis, Rohde et al. found that 92% of the Chinese population experienced unhealthy air (US EPA standards) for more than 120 hours, while 38% experienced an unhealthy average concentration. The population-weighted average $PM_{2.5}$ exposure in China is 52 μg/m$^3$. The observed air pollution causes 1.6 million deaths in China each year, accounting for approximately 17% of all deaths in China during 2014 [20]. In addition, we found that, despite Altay (10.7 μg/m$^3$) being the city with the lowest concentration, the Altay region did not reach the annual average concentration of the WHO Air Quality Guidelines (AQG, 0–10 μg/m$^3$).

**AQI.** From 2015 to 2019, the national annual mean AQI was 72.9, from a regional perspective, Xinjiang (101.4), the Northern region (94.5), the Weihe River Basin (78.7), and the Central region (73.1) were the regions with the highest annual mean from 2015 to 2019 (Table 1). The AQI in Xinjiang was 39% higher than the national annual mean, and Hetian (194.8), Kashi (179.1), Akesu (142.1), Kezhou (135.3), and Tulufan (119.2) in Xinjiang are the five cities with the highest annual mean in the country. Xingtai (119.2), Baoding (118.7), Anyang (118.5), Handan (118.2), and Shijiazhuang (116.6) in the Northern region are the cities with the highest annual mean.

Our results are consistent with the literature. For example, Zhan et al. showed that the AQI levels are higher in Northern China and Xinjiang Province and lower in Southern China, with

**Table 1. Annual mean concentration of pollutants from 2015 to 2019.**

| Regions | PM2.5 | AQI | PM10 | SO2 | CO | NO2 | O3 |
|---------|-------|-----|------|-----|------|-----|------|
| N | 58.2 | 94.5 | 107.2 | 26.9 | 1.17 | 36.6 | 67.4 |
| X | 52.5 | 101.4 | 136.8 | 13.0 | 1.14 | 27.1 | 61.8 |
| E | 40.2 | 67.5 | 68.0 | 13.3 | 0.82 | 32.7 | 63.0 |
| S | 31.4 | 52.4 | 50.4 | 11.3 | 0.86 | 22.9 | 55.2 |
| SC | 40.1 | 65.6 | 66.0 | 13.4 | 0.81 | 28.1 | 53.6 |
| NE | 38.4 | 66.9 | 67.3 | 20.5 | 0.86 | 25.3 | 59.8 |
| W | 41.1 | 78.7 | 92.0 | 20.2 | 0.93 | 29.6 | 63.1 |
| YG | 25.9 | 47.9 | 46.2 | 13.3 | 0.76 | 17.7 | 53.9 |
| C | 46.6 | 73.1 | 75.6 | 15.8 | 0.96 | 27.4 | 59.2 |
| Q | 29.6 | 63.6 | 68.5 | 16.0 | 0.81 | 19.6 | 72.9 |
| IM | 31.8 | 67.5 | 75.0 | 20.8 | 0.77 | 23.8 | 68.7 |
| CH | 42.7 | 72.9 | 78.6 | 17.8 | 0.93 | 28.4 | 61.4 |

$PM_{2.5}$, $PM_{10}$, $SO_2$, CO, $NO_2$, $O_3$: unit: μg/m$^3$, and CO: unit: mg/m$^3$. Regions are identified by the first capital letter: N: Northern; X: Xinjiang; E: Eastern (Yangtze River Delta); S: Southern (Pearl River Delta); SC: Sichuan basins; NE: Northeast; W: Weihe River Basin; YG: Yungui; C: Central; Q: Qinghai-Tibet; IM: Inner Mongolia; and CH: China.

spatial clustering [23]. Fang et al. indicated that urbanization has played an important negative role in determining air quality in Chinese cities. The population, urbanization rate, automobile density, and secondary industry proportion were all found to have had a significant influence over air quality [24]. Wang et al. showed that three dust source areas, namely, 'Northwesterly Sources', 'Northerly Sources', and 'Loess Plateau Source', and an anthropogenic 'Southerly Source' contributed to the high particulate matter concentrations at Xian of the Weihe River Basin [25].

Inner Mongolia (67.5), the Eastern region (67.5), the Northeast region (66.9), the Sichuan Basin (65.6), and Qinghai-Tibet (63.6) (sorted by annual mean) have lower AQI values than the national annual mean (72.9); Yungui (47.9) and Southern (52.4) have the lowest annual mean values. Diqing (34.4) and Lijiang (39.3) in Yungui and Sanya (33.9), Haikou (38.2), and Shenzhen (49.5) in the Southern region are the cities with lowest annual mean. This finding is consistent with a previous study [26].

**SO$_2$.** From 2015 to 2019, the national annual mean SO$_2$ was 17.8 μg/m$^3$, from a regional perspective, the Northern region (26.9 μg/m$^3$), Inner Mongolia (20.8 μg/m$^3$), the Northeast region (20.5 μg/m$^3$), and the Weihe River Basin (20.2 μg/m$^3$) are the regions with the highest annual mean from 2015 to 2019 (Table 1). The value in the Northern region was 51% higher than the national annual mean. Although the annual mean in Inner Mongolia was lower than that in the Northern region, it was still 17% higher than the national annual mean, indicating that the SO$_2$ pollution in the Northern region is serious. Jinzhong (60.5 μg/m$^3$) (in Shanxi Province) in the Northern region is the city with the highest annual mean in the country. Linfen (59.8 μg/m$^3$), Lvliang (53.0 μg/m$^3$) (in Shanxi Province), Xingtai (40.0 μg/m$^3$), and Tangshan (37.6 μg/m$^3$) in the Northern region; Wuhai (47.0 μg/m$^3$), Baotou (30.0 μg/m$^3$), and Hohhot (24.7 μg/m$^3$) in Inner Mongolia; Jinzhou (43.6 μg/m$^3$) and Huludao (39.8 μg/m$^3$) in the Northeast region; and Shizuishan (50.4 μg/m$^3$) and Yinchuan (43.8 μg/m$^3$) in the Weihe River Basin are the cities with the highest annual mean.

Our results are consistent with the literature. For example, Zhang et al. also showed that SO$_2$ pollution is the most serious in Wuhai in Inner Mongolia, followed by Hohhot and Baotou, and the lightest is in Ordos and Bayannur [27].

Qinghai-Tibet (16.0 μg/m$^3$) and the Central region (15.8 μg/m$^3$) have slightly lower values than the national annual mean (17.8 μg/m$^3$). The SO$_2$ concentrations in Yungui (13.3 μg/m$^3$), the Sichuan Basin (13.4 μg/m$^3$), the Eastern region (13.3 μg/m$^3$), and Xinjiang (13.0 μg/m$^3$) are similar; the Southern region (11.3 μg/m$^3$) has the lowest annual mean. Sanya (2.9 μg/m$^3$) and Haikou (5.0 μg/m$^3$) in the Southern region, Fuzhou (5.5 μg/m$^3$) and Ningde (6.0 μg/m$^3$) in the Eastern region, Dali (6.5 μg/m$^3$) in Yungui, and Bazhong (4.4 μg/m$^3$) in the Sichuan Basin are the cities with low annual mean values. Our findings concur with the results showing the highest concentration distribution of SO$_2$ in Shanxi and Hebei reported by Silver et al. [17].

**CO.** From 2015 to 2019, the national annual mean CO was 0.93 mg/m$^3$, from a regional perspective, the Northern region (1.17 mg/m$^3$), Xinjiang (1.14 mg/m$^3$), the Central region (0.96 mg/m$^3$), and the Weihe River Basin (0.93 mg/m$^3$) are the regions with the highest annual mean from 2015 to 2019 (Table 1). The value in the Northern region was 26% higher than the national annual mean. Linfen (2.16 mg/m$^3$) and Tangshan (2.00 mg/m$^3$) in the Northern region are the cities with the highest annual mean in the country. Yilihasake (1.76 mg/m$^3$) in Xinjiang is the city with the highest annual mean.

The Southern region (0.86 mg/m$^3$), Northeast region (0.86 mg/m$^3$), Eastern region (0.82 mg/m$^3$), Sichuan Basin (0.81 mg/m$^3$), and Qinghai-Tibet (0.81 mg/m$^3$) have lower values than the national annual mean (0.93 mg/m$^3$); the annual means in Yungui (0.76 mg/m$^3$) and Inner Mongolia (0.77 mg/m$^3$) were the lowest. Sanya (0.60 mg/m$^3$), and Haikou (0.61 mg/m$^3$) in the

Southern region, Fuzhou (0.68 mg/m$^3$) and Ningde (0.80 mg/m$^3$) in the Eastern region, Dali (0.67 mg/m$^3$) in Yungui, and Hulunbeier in Inner Mongolia are the cities with lowest annual mean.

**NO$_2$.** From 2015 to 2019, the national annual mean NO$_2$ was 28.4 μg/m$^3$, from a regional perspective, the Northern region (36.6 μg/m$^3$), Weihe River Basin (29.6 μg/m$^3$), and Eastern region (32.7 μg/m$^3$) were the regions with the highest annual mean from 2015 to 2019 (Table 1). The value in the Northern region was 26% higher than the national annual mean. Tangshan (55.9 μg/m$^3$) and Xingtai (53.2 μg/m$^3$) in the Northern region are the cities with the highest annual mean in the country. Xian (50.0 μg/m$^3$) in the Weihe River Basin and Huzhou (49.3 μg/m$^3$) and Suzhou (46.2 μg/m$^3$) in the Eastern region are the cities with the highest annual mean values.

The fast developing resource and pollution intensive industries along with the 'Go West' movement and weak emission controls [28] contributed to the higher rate of increase in NO$_2$ over the Western region from 2005–2013 than over that over the Southwestern, Northern, Eastern, and Southern regions.

The annual mean values in the Sichuan Basin (28.1 μg/m$^3$), Xinjiang (27.1 μg/m$^3$), and the Central region (27.4 μg/m$^3$) are similar, the values in the Northeast region (25.3 μg/m$^3$), Inner Mongolia (23.8 μg/m$^3$), and the Southern region (22.9 μg/m$^3$) are lower than the national annual mean. Chengdu (45.0 μg/m$^3$) has the highest annual mean in the Sichuan Basin, Yungui (17.7 μg/m$^3$) and Qinghai-Tibet (19.6 μg/m$^3$) have the lowest annual mean, Lijiang (11.3 μg/m$^3$) in Yungui and Linzhi in Qinghai-Tibet are the cities with the lowest annual mean. Silver et al. also found higher concentrations of NO$_2$ in Tianjin, Hebei, Beijing, and Shanghai in the east and Hong Kong and Chongqing in the Sichuan Basin [17].

We found that NO$_2$ shows a different trend in cities in the same region. The concentration in certain cities has increased, while the concentration in nearby cities has decreased, although they are in the same region (Fig 1). Krotkov et al. also found that NO$_2$ has a large spatial heterogeneity [9]. Silver et al. suggested that the trend of spatial heterogeneity of NO$_2$ may be partly due to its relatively short lifespan [17].

**O$_3$.** From 2015 to 2019, the national annual mean O$_3$ was 61.4 μg/m$^3$, from a regional perspective, Qinghai-Tibet (72.9 μg/m$^3$), Inner Mongolia (68.7 μg/m$^3$), the Northern region (67.4 μg/m$^3$), the Weihe River Basin (63.1 μg/m$^3$), the Eastern region (63.0 μg/m$^3$), and Xinjiang (61.8 μg/m$^3$) had a higher annual mean (median) concentration from 2015 to 2019 (Table 1). The value in Qinghai-Tibet was 9% higher than the national annual mean. Lhasa, Haibeizhou, Haixizhou, and Guoluozhou in Qinghai-Tibet and Alashanmeng in Inner Mongolia are the cities with the highest annual mean in the country. High O$_3$ concentrations in the city of Lhasa of Tibet may be associated with stronger photochemical reactions, vertical mixing and downward transport of stratospheric air mass.

The annual mean values of the Northeast region (59.8 μg/m$^3$) and the Central region (59.2 μg/m$^3$) were similar and slightly lower than the national annual mean (61.4 μg/m$^3$). The Southern region (55.2 μg/m$^3$), Yungui (53.9 μg/m$^3$), and the Sichuan Basin (53.6 μg/m$^3$) had the lowest annual mean values. Chengdu (45.0 μg/m$^3$) had the highest annual mean in the Sichuan Basin. Chongqing (41.5 μg/m$^3$) and Bazhong (44.1 μg/m$^3$) in the Sichuan Basin and Xishuangbanna (37.9 μg/m$^3$), Nujiang (42.2 μg/m$^3$), and Dehong (44.7 μg/m$^3$) in Yungui were the cities with the lowest annual mean.

Silver et al. also found that the highest concentrations of the O$_3$ were in high-altitude provinces in Tibet and Qinghai and Hong Kong, while Chongqing had the lowest O$_3$ concentration [17]. Obviously, the spatial distribution of O$_3$ is different from the spatial distribution of the above pollutants.

## Changes over time

**PM$_{2.5}$.** Nationally, the annual average relative change was -7.4% and annual average relative change of 93% cities was negative from 2015 to 2019. The annual mean of PM$_{2.5}$ decreased continuously year by year. The annual mean in 2019 was 27.9% (relative change) lower than that in 2015. Compared with 2015, the annual mean of 350 cities decreased in 2019 (Tables 2–5; S1–S9 Tables in S1 File; Figs 1 and 2). The number of cities where PM$_{2.5}$ meets the air quality guidelines of WHO (annual mean 0–10 μg/m$^3$) has increased from 0 in 2015 to 3 in 2019. The number of cities where PM$_{2.5}$ reached target 1 of the transition period of WHO (annual mean 25–35 μg/m$^3$) has increased from 53 in 2015 to 119 in 2019. In comparison with 2017 [18], there are more cities which meets the WHO guidelines and the WHO target 1 transition period (annual mean 25–35 μg/m$^3$). Our results are comparable with literature. For example, Zheng et al. found that since the "Action Plan" was implemented, the annual mean of national population-weighted PM$_{2.5}$ has decreased by 21.5%, from 60.5 μg/m$^3$ in 2013 to 47.5 μg/m$^3$ in 2015 [29]. Silver et al. found that the decline rate of PM$_{2.5}$ was maintained between 2015 and

**Table 2. Annual average relative change (%) of pollutants from 2015 to 2019, regions are indicated by the first capital letter.**

| Regions | PM$_{2.5}$ | AQI | PM$_{10}$ | SO$_2$ | CO | NO$_2$ | O$_3$ |
|---|---|---|---|---|---|---|---|
| N | -7.0 | -4.7 | -5.6 | -21.8 | -9.2 | -2.5 | 6.7 |
| X | -3.0 | -2.1 | -0.7 | -13.3 | -8.5 | 0.7 | 7.0 |
| E | -8.4 | -6.3 | -6.8 | -17.5 | -5.8 | -3.8 | 4.0 |
| S | -6.5 | -4.7 | -4.8 | -8.9 | -4.8 | -1.5 | 2.4 |
| SC | -7.5 | -6.9 | -9.1 | -13.5 | -5.7 | -1.6 | 3.2 |
| NE | -8.7 | -6.6 | -7.0 | -16.8 | -6.8 | -6.2 | 2.5 |
| W | -7.3 | -5.1 | -5.9 | -16.0 | -10.6 | -1.3 | 4.2 |
| YG | -4.9 | -4.1 | -5.6 | -12.5 | -3.8 | -1.9 | 3.8 |
| C | -6.8 | -4.7 | -5.6 | -17.8 | -4.9 | -1.5 | 7.0 |
| Q | -16.3 | -9.4 | -14.1 | -11.8 | -3.7 | -1.4 | 5.8 |
| IM | -9.9 | -6.7 | -7.3 | -14.3 | -8.3 | -2.5 | 3.2 |
| CH | -7.4 | -5.3 | -5.9 | -16.3 | -6.7 | -2.5 | 4.8 |

**Table 3. Relative change (%) of pollutants between 2015 and 2019, regions are indicated by the first capital letter.**

| Regions | PM$_{2.5}$ | AQI | PM$_{10}$ | SO$_2$ | CO | NO$_2$ | O$_3$ |
|---|---|---|---|---|---|---|---|
| N | -26.6 | -17.8 | -20.5 | -63.9 | -32.0 | -9.9 | 29.0 |
| X | -17.5 | -10.7 | -11.0 | -42.9 | -33.5 | -0.1 | 26.5 |
| E | -30.3 | -23.3 | -25.2 | -54.5 | -21.8 | -15.6 | 13.2 |
| S | -24.8 | -18.6 | -18.9 | -32.2 | -19.5 | -7.3 | 9.5 |
| SC | -28.3 | -25.8 | -33.0 | -44.8 | -22.7 | -9.1 | 10.7 |
| NE | -33.0 | -24.9 | -27.0 | -53.8 | -26.4 | -23.5 | 8.3 |
| W | -27.7 | -20.7 | -24.7 | -50.9 | -37.2 | -5.5 | 15.2 |
| YG | -20.6 | -16.3 | -22.5 | -40.1 | -16.7 | -8.5 | 14.1 |
| C | -26.2 | -18.2 | -21.2 | -55.7 | -18.8 | -7.9 | 29.5 |
| Q | -52.7 | -34.9 | -52.2 | -42.6 | -21.9 | -7.1 | 7.1 |
| IM | -35.8 | -25.7 | -30.1 | -47.3 | -30.0 | -12.2 | 12.1 |
| CH | -27.9 | -20.5 | -23.8 | -51.2 | -25.3 | -10.6 | 18.3 |

**Table 4. Quantity of cities showing '< 0' and '> 0' of relative change between 2015 and 2019, regions are indicated by the first capital letter.**

| Regions | $PM_{2.5}$ | | $PM_{10}$ | | AQI | | $SO_2$ | | CO | | $NO_2$ | | $O_3$ | |
|---|---|---|---|---|---|---|---|---|---|---|---|---|---|---|
| | < 0 | > 0 | < 0 | > 0 | < 0 | > 0 | < 0 | > 0 | < 0 | > 0 | < 0 | > 0 | < 0 | > 0 |
| N | 68 | 3 | 66 | 5 | 66 | 5 | 71 | 0 | 66 | 5 | 53 | 18 | 6 | 65 |
| X | 14 | 2 | 14 | 2 | 14 | 2 | 16 | 0 | 16 | 0 | 10 | 6 | 5 | 11 |
| E | 48 | 0 | 48 | 0 | 48 | 0 | 47 | 1 | 47 | 1 | 43 | 5 | 9 | 39 |
| S | 37 | 0 | 37 | 0 | 37 | 0 | 34 | 3 | 37 | 0 | 29 | 8 | 12 | 25 |
| SC | 21 | 1 | 22 | 0 | 22 | 0 | 20 | 2 | 20 | 2 | 18 | 4 | 7 | 15 |
| NE | 37 | 1 | 38 | 0 | 38 | 0 | 38 | 0 | 35 | 3 | 37 | 1 | 11 | 27 |
| W | 29 | 0 | 29 | 0 | 29 | 0 | 29 | 0 | 28 | 1 | 21 | 8 | 5 | 24 |
| YG | 20 | 4 | 21 | 3 | 21 | 3 | 22 | 2 | 20 | 4 | 21 | 3 | 6 | 18 |
| C | 54 | 1 | 52 | 3 | 52 | 3 | 55 | 0 | 51 | 4 | 43 | 12 | 2 | 53 |
| Q | 11 | 0 | 11 | 0 | 11 | 0 | 11 | 0 | 9 | 2 | 7 | 4 | 3 | 8 |
| IM | 11 | 0 | 11 | 0 | 11 | 0 | 11 | 0 | 10 | 1 | 7 | 4 | 1 | 10 |
| CH | 350 | 12 | 349 | 13 | 349 | 13 | 354 | 8 | 339 | 23 | 289 | 73 | 67 | 295 |

**Table 5. Quantity of cities showing '< 0' and '> 0' of annual average relative change from 2015 to 2019, regions are indicated by the first capital letter.**

| Regions | $PM_{2.5}$ | | $PM_{10}$ | | AQI | | $SO_2$ | | CO | | $NO_2$ | | $O_3$ | |
|---|---|---|---|---|---|---|---|---|---|---|---|---|---|---|
| | < 0 | > 0 | < 0 | > 0 | < 0 | > 0 | < 0 | > 0 | < 0 | > 0 | < 0 | > 0 | < 0 | > 0 |
| N | 66 | 5 | 65 | 6 | 65 | 6 | 71 | 0 | 65 | 6 | 51 | 20 | 5 | 66 |
| X | 10 | 6 | 14 | 2 | 14 | 2 | 16 | 0 | 16 | 0 | 10 | 6 | 3 | 13 |
| E | 48 | 0 | 48 | 0 | 48 | 0 | 46 | 2 | 47 | 1 | 42 | 6 | 7 | 41 |
| S | 37 | 0 | 37 | 0 | 37 | 0 | 34 | 3 | 35 | 2 | 27 | 10 | 10 | 27 |
| SC | 21 | 1 | 22 | 0 | 22 | 0 | 20 | 2 | 18 | 4 | 14 | 8 | 6 | 16 |
| NE | 36 | 2 | 38 | 0 | 38 | 0 | 37 | 1 | 35 | 3 | 36 | 2 | 9 | 29 |
| W | 27 | 2 | 27 | 2 | 27 | 2 | 29 | 0 | 28 | 1 | 21 | 8 | 3 | 26 |
| YG | 18 | 6 | 21 | 3 | 21 | 3 | 22 | 2 | 20 | 4 | 18 | 6 | 5 | 19 |
| C | 51 | 4 | 52 | 3 | 52 | 3 | 53 | 2 | 50 | 5 | 41 | 14 | 2 | 53 |
| Q | 11 | 0 | 11 | 0 | 11 | 0 | 10 | 1 | 7 | 4 | 7 | 4 | 3 | 8 |
| IM | 11 | 0 | 11 | 0 | 11 | 0 | 11 | 0 | 9 | 2 | 7 | 4 | 1 | 10 |
| CH | 336 | 26 | 346 | 16 | 346 | 16 | 349 | 13 | 330 | 32 | 274 | 88 | 54 | 308 |

2017. Among the 1689 monitoring stations in China, 58.4% have undergone a significant relative change in $PM_{2.5}$, 90% of which are negative [17]. Lin et al. used satellite data to suggest that the Chinese $PM_{2.5}$ trend steepened from -0.65 μg/m$^3$ year$^{-1}$ between 2006–2010 to -2.3 μg/m$^3$ year$^{-1}$ between 2011 and 2015 [30]. The Statistical Communique of the People's Republic of China (SCPRC) shows that the total amount of particulate emissions decreased year by year from 15.38 million tons to 7.96 million tons from 2015 to 2017, and it decreased by 48.2% from 2015 to 2017, which is the key factor for the downward trend of $PM_{2.5}$ ($PM_{10}$) [31–33].

From a regional perspective, the relative change in the annual mean (median) of $PM_{2.5}$ in all regions from 2015 to 2019 was negative. Qinghai-Tibet (-16.3%), Inner Mongolia (-9.9%), the Northeast region (-8.7%), the Eastern region (-8.4%), and the Sichuan Basin (-7.5%) are the regions with a rapid decline (sorted by relative change). The average change in Qinghai-Tibet was 120% higher than the national relative change. Ali (-20%) of the Qinghai-Tibet, Hegang (-33.9%) and Baicheng (-18.8%) in the Northeast region, and Haimen (-14.4%) and Jinhua (-12.7%) in the Eastern region are the cities with a faster decline.

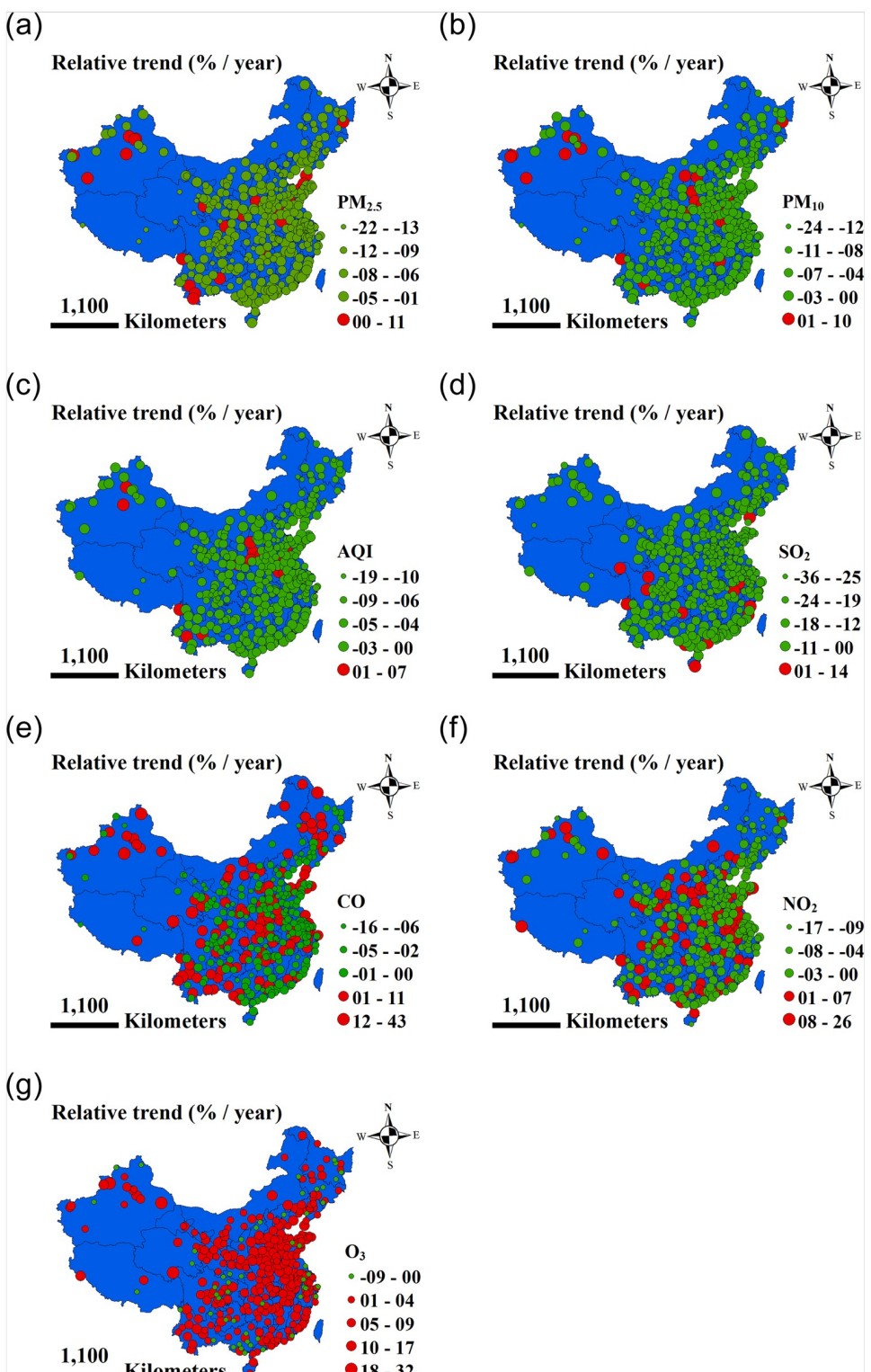

**Fig 1. Sign of the annual average relative change (trend, %) from 2015 to 2019 (green: Significantly negative, and red: Significantly positive).**

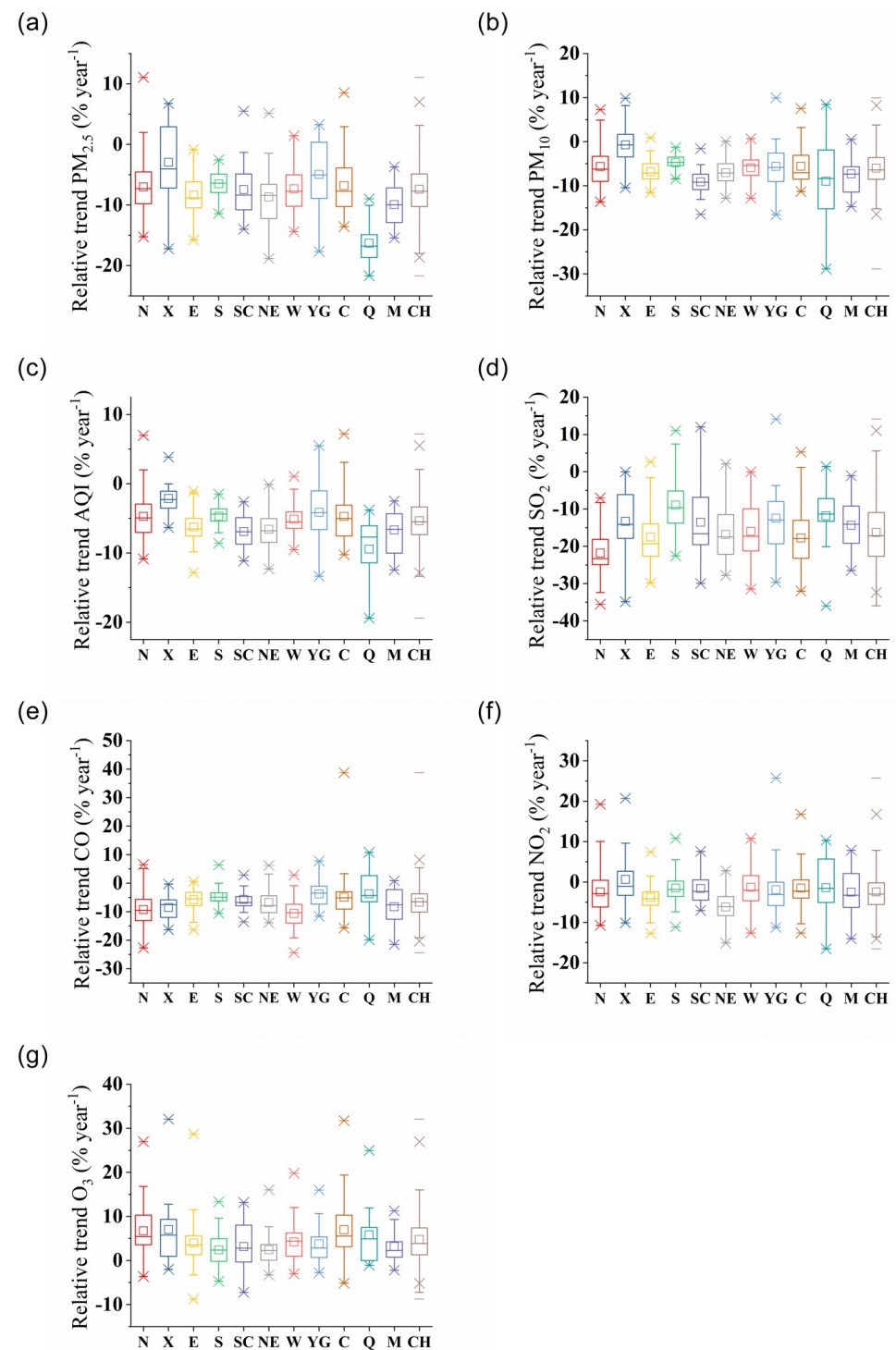

**Fig 2. Annual average relative change from 2015 to 2019 in pollutants by region.** The mean (□), median (—), 1% (×). 99% (×), maximum (—), minimum (—), and interquartile range (IQR) of the trend in each region is shown.

Yin et al. showed that the 'Lhasa pattern' of Qinghai-Tibet may serve as a positive example for other regional hub cities. Effective air pollution control measures collectively contributed to the synchronous improvement of the economy and air quality in Lhasa, moreover, lower concentrations of air pollutants are observed in Lhasa except for $O_3$ because of the relatively isolated location, low air pollutant emissions associated with its industrial structure and renewable energy consumption [34]. Qiu et al. showed that in Baotou, which is a typical industrial city in Inner Mongolia for evaluating the current national control measures, that the total emissions of $SO_2$, $NO_X$, and $PM_{2.5}$ were 211.2 Gg, 156.1 Gg, and 28.8 Gg in 2013, respectively, and should be reduced to 39.0%, 32.0%, and 24.4% in 2020, respectively. Even for a typical industrial city, the reduction of $PM_{2.5}$ concentrations not only requires decreases in emissions from the industrial sector as well as residential sources [35]. Zheng et al. showed that the emission reduction rates markedly accelerated after the year 2013, thus confirming the effectiveness of China's Clean Air Action policy. From 2013–2017, China's anthropogenic emissions decreased by 59 % for $SO_2$, 21 % for NOx, 23 % for CO, 36 % for $PM_{10}$, and 33 % for $PM_{2.5}$. Emission control measures are the main drivers of this reduction, and pollution controls on power plants and industries are the most effective mitigation measures [36].

The Weihe River Basin (-7.3%), the Northern region (-7.0%), the Central region (-6.8%), and the Southern region (-6.5%) have slightly lower values than the national relative change. Yungui has second lowest relative change (-4.9%) because it has the lowest concentration in the country, and there is little room for a decrease. Xinjiang has the lowest relative change (-3.0%), which is 59% lower than the national relative change, but Xinjiang has second highest concentration in the country. Xinjiang has the characteristics of high pollution and low improvement. This is at least partially due to the natural PM sources, such as dust. Lu et al. showed that the high $PM_{2.5}$ concentration was mainly affected by sand and dust in the northwest of China and by human activities in the eastern region [37]. Zhang et al. showed that in western China, dust particles are very important for $PM_{2.5}$ and the current control strategy of $PM_{2.5}$ (that is, reducing VOC and PM emissions from fossil / non-fossil combustion) will only partially reduce the pollution of $PM_{2.5}$ of the western region [15]. Cai et al. noted that with the implementation of the "Action Plan", the emissions of $SO_2$, $NO_X$, and $PM_{2.5}$ will decrease by 40%, 44%, and 40% in 2020, from the 2012 levels in Jing-Jin-Ji, respectively. Consequently, the ambient annual $PM_{2.5}$ concentration of 2020 will be 37.8% lower than that in 2012. Thus, the "Action Plan" provided an effective approach for alleviating $PM_{2.5}$ pollution levels in the Jing-Jin-Ji region [3].

From 2015 to 2019, the relative changes in the three major economic belts in the Northern, Eastern (Yangtze River Delta), and Southern regions (Pearl River Delta) are -26.6%, -30.3%, and -24.8%, respectively. Zhang et al. also noted that national and regional concentrations declined in all years from 2013 to 2017, the $PM_{2.5}$ of the Northern region (Beijing, Tianjin, Hebei and the surrounding areas), the Yangtze River Delta and the Pearl River Delta have decreased by 38%, 27%, and 21%, respectively [2]. Silver et al. found that the relative changes in the median $PM_{2.5}$ in all provinces except Shanxi and Jiangxi from 2015 to 2017 showed a negative trend [17]. At the same time, Lin et al. also estimated consistent trends in satellite data from 2011 to 2015 [30]. Jiang et al. estimated that the "Action Plan" would decrease the $PM_{2.5}$ by 26%, $SO_2$ by 34%, and $NO_2$ by 28% in the Pearl River Delta [4].

**$PM_{10}$.** Nationally, the annual average relative change was -5.9% and the annual average relative change of 96% of cities was negative from 2015 to 2019. The annual mean $PM_{10}$ decreased continuously year by year. The annual mean in 2019 was 23.8% (relative change) lower than that in 2015, compared with 2015, the annual mean of 349 cities decreased in 2019 (Tables 2–5; S10–S18 Tables in S1 File; Figs 1 and 2). The number of cities where $PM_{10}$ meets the air quality guidelines of WHO (annual mean 20 μg/m$^3$) increased from 0 in 2015 to 3 in

2019; the number of cities where $PM_{10}$ reached target 3 of the transition period of WHO (annual mean 20–30 μg/m$^3$) increased from 1 in 2015 to 8 in 2019; the number of cities where $PM_{10}$ reached target 1 of the transition period of WHO (annual mean < 70 μg/m$^3$) increased from 117 in 2015 to 229 in 2019. Guo et al. reported that the number of cities where $PM_{10}$ reached target 3 of the transition period of WHO (annual mean 20–30 μg/m$^3$) increased from 3 in 2015 to 8 in 2017. The number of cities where $PM_{10}$ reached target 1 of the transition period of WHO (annual mean < 70 μg/m$^3$) increased from 123 in 2015 to 182 in 2017, compared with 2015, the annual mean of 309 cities decreased in 2017 [18].

From a regional perspective, the relative change in the annual mean (median) $PM_{10}$ in all regions from 2015 to 2019 was negative. Qinghai-Tibet (-14.1%), the Sichuan Basin (-9.1%), Inner Mongolia (-7.3%), the Northeast region (-7.0%), and the Eastern region (-6.8%) are the regions with a rapid decline (sorted by relative change). The value in Qinghai-Tibet was 139% higher than the national relative change. Naqu (-24.1%) and Guoluo (-23.7%) in Qinghai-Tibet, Chengdu (-11.6%) and Zigong (-10.7%) in the Sichuan Basin, and Hulunbeier (-14.7%) in Inner Mongolia are the cities with the fasted decline.

The values in the Weihe River Basin (-5.9%), the Northern region (-5.6%), the Central region (-5.6%), and Yungui (-5.6%) are equal to or slightly lower than the national relative change (-5.9%). The Southern region has a lower relative change (-4.8%) because it has the lower concentration in the country and, thus, there is not much room for a decrease. Xinjiang has the lowest relative change (-0.7%), which is 88% lower than the national relative change, but Xinjiang has the highest concentration in the country. Xinjiang has the characteristics of high pollution and low improvement.

From 2015 to 2019, the relative changes in the three major economic belts in the Northern, Eastern (Yangtze River Delta), and Southern regions (Pearl River Delta) were -20.5%, -25.2%, and -18.9%, respectively. Beijing, Shanghai, Guangzhou, and Shenzhen have values of -11.0%, -10.3%, -5.1%, and -4.5%, respectively, no city exceeded the national relative change of -23.8%.

**AQI.** Nationally, the annual average relative change was -5.3% and the annual average relative change of 96% cities was negative from 2015 to 2019. The annual mean AQI decreased continuously year by year. The annual mean in 2019 was 20.5% (relative change) lower than that in 2015. The annual mean of 349 cities decreased in 2019 (Tables 2–5; S19–S27 Tables in S1 File; Figs 1 and 2).

From a regional perspective, the relative change in the annual mean (median) AQI in all regions from 2015 to 2019 was negative. Qinghai-Tibet (-9.4%), the Sichuan Basin (-6.9%), Inner Mongolia (-6.7%), the Northeast region (-6.6%), and the Eastern region (-6.3%) are the regions with a rapid decline (sorted by relative change). The value in Qinghai-Tibet was 74% higher than the national relative change.

The values in the Weihe River Basin (-5.1%), the Northern region (-4.7%), the Central region (-4.7%), and the Southern region (-4.7%) are slightly lower than the national relative change (-5.3%). Yungui has the lowest relative change (-4.1%) because it has the lowest concentration in the country and there is little room for a decrease. Xinjiang has the lowest relative change (-2.1%), which is 60% lower than the national relative change. Xinjiang also has the highest concentration in the country. Xinjiang has the characteristics of high pollution and low improvement.

From 2015 to 2019, the relative changes in the three major economic belts in the Northern region, the Yangtze River Delta, and the Pearl River Delta were -17.8%, -23.3%, and -18.6%, respectively. Beijing, Shanghai, Guangzhou, and Shenzhen exhibited changes of -36.7%, -29.7%, -18.9%, and -14.8%, respectively. Beijing and Shanghai exceeded the national relative change of -20.5%.

**SO$_2$.** Nationally, the annual average relative change was -16.3% and the annual average relative change of 96% cities was negative from 2015 to 2019. The annual mean SO$_2$ decreased continuously year by year. Silver et al. also found that the annual average relative change of 90% of monitoring stations was negative from 2015 to 2017 [17]. The annual mean in 2019 was 51.2% (relative change) lower than that in 2015. Compared with 2015, the annual mean of 354 cities decreased in 2019 (Tables 2–5; S28–S36 Tables in S1 File; Figs 1 and 2), the number of cities where SO$_2$ meets limit of Level 1 of the ambient air quality standard of China (GB 3095–2012) (annual mean 0–20 μg/m$^3$) increased from 157 in 2015 to 342 in 2019. Guo et al. reported that number of cities where SO$_2$ meets limit of Level 1 of the ambient air quality standard of China (GB 3095–2012) (annual mean 0–20 μg/m$^3$) increased from 153 in 2015 to 268 in 2017 [18].

From a regional perspective, the annual average relative changes in the mean (median) in all regions from 2015 to 2019 were negative. The Northern (-21.8%), Central (-17.8%), Eastern (-17.5%), and Northeast regions (-16.8%) were the regions with a rapid decline (sorted by relative change). The change in the Northern region was 34% higher than the national annual average relative change. Zhumadian (-35.5%), Sanmenxia (-32.4%) in the Northern region, Fuyang (-32.0%), Yiyang (-32.0%), and Bozhou (-31.5%) in the Central region, Zhuji (-29.8%) and Shaoxing (-29.4%) in the Eastern region, and Tonghua (-27.6%) and Mudanjiang (-27.1%) in the Northeast region were the cities with the fastest decline.

Huang et al. reported that the largest reductions in annual sulphur dioxide and carbon monoxide concentrations occurred in the Beijing-Tianjin-Hebei region from 2013 to 2017 [38].

The Weihe River Basin (-16.0%), Inner Mongolia (-14.3%), the Sichuan Basin (-13.5%), and Xinjiang (-13.3%) had slightly lower values than the national relative change (-16.3%). Yungui (-12.5%), Qinghai-Tibet (-11.8%) and the Southern region (-8.9%) had the lowest relative change because these regions have almost the lowest concentrations in the country, thus, there is little room for a decrease. Compared with other pollutants, it can be seen that the SO$_2$ in all regions of the country has decreased significantly, the minimum value is also -8.9%, and all regions have greatly improved.

From 2015 to 2019, the relative changes in the three major economic belts in the Northern region, the Yangtze River Delta, and the Pearl River Delta were -63.9%, -54.5%, and -32.2%, respectively. Huang et al. also reported that in the Beijing-Tianjin-Hebei region, reductions of 63.5% occurred for sulphur dioxide and 30.5% occurred for carbon monoxide from 2013 to 2017 [38].

**CO.** Nationally, the annual average relative change was -6.7% and annual average relative change of 91% cities was negative from 2015 to 2019. The annual mean CO value decreased continuously year by year (Tables 2–5; S37–S45 Tables in S1 File; Figs 1 and 2). The number of cities where CO meets Level 1 of the ambient air quality standard of China (GB 3095–2012) (annual mean 0–40 μg/m$^3$) increased from 180 in 2015 to 328 in 2019. Guo et al. also reported that the number of cities where CO meets Level 1 of ambient air quality standard of China (GB 3095–2012) (annual mean 0–40 μg/m$^3$) increased from 181 in 2015 to 241 in 2017 [18].

From a regional perspective, annual average relative changes in the mean (median) values in all regions from 2015 to 2019 were negative. The Weihe River Basin (-10.6%), the Northern region (-9.2%), Xinjiang (-8.5%), Inner Mongolia (-8.3%), and the Northeast region (-6.8%) were the regions with a rapid decline (sorted by relative change). The Weihe River Basin was 58% higher than the national annual average relative change. Weiwu (-24.4%) and Shangluo (-19.2%) in the Weihe River Basin, Lvliang (-22.7%) and Dezhou (-20.3%) in the Northern region, Aletai (-16.3%) and Hetian (-14.7%) in Xinjiang, and Xilinguolemeng (-21.5%) and Alashanmeng (-14.2%) in Inner Mongolia were the cities with the fastest decline.

The values in the Eastern region (-5.8%), Sichuan Basin (-5.7%), Central region (-4.9%), and Southern region (-4.8%) were slightly lower than the national relative change (-6.7%).

Yungui (-3.8%), Qinghai-Tibet (-3.7%) had the lowest relative change because it has the almost lowest concentration in the country, thus, there is little room for a decrease.

From 2015 to 2019, the relative changes in the three major economic belts in the Northern region, the Yangtze River Delta, and the Pearl River Delta were -32.0%, -21.8%, and -19.5%, respectively.

Huang et al. reported that in the Beijing-Tianjin-Hebei region, reductions of 30.5% occurred for carbon monoxide from 2013 to 2017 [38]. Streets et al. showed that emissions of carbon monoxide are projected to decline from 115 mt in 1995 to 96.8 mt in 2020 due to more efficient combustion techniques, especially in the transportation sector, although if these measures are not realized, carbon monoxide emissions could increase to 130 mt by 2020 [39].

**$NO_2$.** Zhao et al. found a nonlinear relationship between $PM_{2.5}$ and the precursor $NO_2$ and showed that the effects of strengthened vehicle emission standards on national air quality improvement were hindered by the complex nonlinear response of the $PM_{2.5}$ concentration to NOx emissions [40].

Nationally, the annual average relative change was only -2.5% and the annual average relative change of only 76% cities was negative from 2015 to 2019 (Tables 2–5; Tables S46–S54 in S1 File; Figs 1 and 2). From 2017 to 2019, compared with 2015 to 2017, China's $NO_2$ exhibited a significant downward trend; the annual mean (annual average median) values of $NO_2$ in 2015, 2016, and 2017 were 29.2 μg/m$^3$ (28.4 μg/m$^3$), 29.4 μg/m$^3$ (28.4 μg/m$^3$), and 30.3 μg/m$^3$ (29.2 μg/m$^3$), respectively, demonstrating that $NO_2$ pollution did not improve from 2015 to 2017. Silver et al. reported that the annual average relative change in the annual average median $NO_2$ from 2015 to 2017 was 0.1%, and there was no median trend, the relative change of the sites of 46% showed a negative trend, and our results are consistent with the results of Silver et al. [17]. Guo et al. also reported that the annual average $NO_2$ concentration from 2015 to 2017 hardly changed [18]. The number of cities where $NO_2$ meets Level 1 of the ambient air quality standard of China (GB 3095–2012) (annual mean < 40 μg/m$^3$) increased from 294 in 2015 to 338 in 2019. Guo et al. reported that the number of cities where $NO_2$ meets Level 1 of the ambient air quality standard of China (GB 3095–2012) (annual mean < 40 μg/m$^3$) increased from 299 in 2015 to 301 in 2017 [18]. These results are consistent with the trend we found. By 2019, our research found that the $NO_2$ concentration was reduced and pollution was improved.

From a regional perspective, the annual average relative changes in the mean (median) in regions from 2015 to 2019 were negative. The Northeast (only -6.2%) and Eastern (only -3.8%) regions are the regions with a fast decline (sorted by relative change). The value in the Northeast region was 148% higher than the national annual average relative change. Songyuan (-15.1%) and Suihua (-13.6%) in the Northeast region and Shaoxing (-12.7%) and Quzhou (-10.0%) in the Eastern region are the cities with the fastest decline.

The Northern region (-2.5%) and Inner Mongolia (-2.5%) are equal to the national annual average relative change. Yungui (-1.9%), Sichuan Basin (-1.6%), the Central region (-1.5%), the Southern region (-1.5%), and Qinghai-Tibet (-1.4%) have lower values than the national annual average relative change (-2.5%). Xinjiang has a positive annual average relative change (+0.7%) because Hami (+20.7%) in the Xinjiang region is the city with the fastest increasing rate. However, Xinjiang has a high concentration throughout the country. Xinjiang has the characteristics of high pollution and low improvement.

From 2015 to 2019, the relative changes in the three major economic belts in the Northern, the Eastern (Yangtze River Delta), and Southern (Pearl River Delta) regions were -9.9%, -15.6%, and -7.3%, respectively.

Huang et al. reported that a key strength of the Air Pollution Prevention and Control Action Plan (APPCAP) is that it demonstrates China's ability to control air pollution.

However, the emissions control efforts for $O_3$ and $NO_2$ require further attention and should be strengthened because the average annual $O_3$ concentrations increased from 139.2 μg/m$^3$ in 2013 to 162.9 μg/m$^3$ in 2017 in the 74 key cities while average annual $NO_2$ concentrations only decreased from 43.9 μg/m$^3$ in 2013 to 39.2 μg/m$^3$ in 2017 [38].

**$O_3$.** Nationally, the annual average relative change was 4.8% and the annual average relative change of 85% cities was positive from 2015 to 2019. The annual mean in 2019 was 18.3% (relative change) higher than that in 2015. Compared with 2015, the annual mean of 295 cities increased in 2019 (Tables 2–5; S55–S63 Tables in S1 File; Figs 1 and 2), the number of cities with a value greater than 60.0 μg/m$^3$ increased from 123 in 2015 to 235 in 2019, which indicates that $O_3$ should become the new focus in the prevention of air pollution. Verstraeten et al. reported that satellite observations show that from 2005 to 2010, the concentration of China's ozone ($O_3$) steadily increased at a rate of 7% per year [10]. Silver et al. found that in contrast to $PM_{2.5}$ and $SO_2$, the annual mean $O_3$ MDA8 exhibited a positive median trend of 4.6 μg m$^{-3}$ year$^{-1}$ or 5.2% year$^{-1}$. A total of 55% of stations exhibited significant trends, and of these, 92% were positive. Across all stations, the percentage of days where the WHO AQG (100 μg/m$^3$) was exceeded for MDA8 rose from 9.8% in 2015 to 12.4% in 2017. The annual mean $O_3$ values showed similar relative and absolute trends, which is consistent with our results [17].

From a regional perspective, the annual average relative changes in the mean (median) in all regions from 2015 to 2019 were positive. Xinjiang (7.0%), the Central region (7.0%), the Northern region (6.7%), and Qinghai-Tibet (5.8%) were the regions with a rapid increase (sorted by relative change). Hami (22.6%) in Xinjiang and Chuzhou (37.1%) and Wuhu (19.4%) in the Central region were the cities with the fastest rates of increase.

Li et al. reported that a more important factor for ozone trends in the North China Plain is the 40% decrease of $PM_{2.5}$ over the 2013–2017 period, which slowed down the aerosol sink of $HO_2$ radicals and stimulated $O_3$ production [41].

The Weihe River Basin (4.2%), the Eastern region (4.0%), Yungui (3.8%), Inner Mongolia (3.2%), and the Sichuan Basin (3.2%) had lower values than the national relative change (4.8%). The Southern region (2.4%) and the Northeast region (2.5%) had the lowest relative changes.

Li et al. reported that from 2013–2017, increasing ozone trends of 1–3 ppbv a$^{-1}$ occurred in the megacity clusters of eastern China, which we attribute to changes in anthropogenic emissions. Anthropogenic NOx emissions in China are estimated to have decreased by 21%, whereas volatile organic compounds (VOCs) emissions changed little. Decreasing NOx would increase ozone under the VOC-limited conditions thought to prevail in urban China [41].

From 2015 to 2019, the relative changes in the three major economic belts in the Northern region, the Yangtze River Delta, and the Pearl River Delta were 29.0%, 13.2%, and 9.5%, respectively.

## Factors contributing to the decrease in air pollutant levels

A number of policy actions contributed to the decrease in $PM_{2.5}$, $SO_2$, $NO_2$, and CO levels. Firstly, the emission standards of thermal power plants and all emission-intensive industrial sectors (such as steel and cement) have been strengthened. By the end of 2017, more than 95% of China's coal-fired power plants were equipped with flue gas desulfurization (FGD) and selective catalytic reduction (SCR) or selective non-catalytic reduction (SNCR) systems and 71% of the coal-fired power generation capacity reached the "ultra-low emission" standard. In addition, industrial boiler were upgraded and small coal-fired boilers were eliminated, which was important because large-scale operation boilers are widely equipped with $SO_2$ and particulate matter control devices. The elimination of backward industries phases out obsolete or

inefficient technology in various industries and allows for structural adjustments. In addition, clean fuel was promoted in the residential sector and advanced stoves and clean coal nation-wide were promoted from 2013 to 2016. In 2017, the use of natural gas and electricity to replace coal was further promoted, which affected 6 million households nationwide, of which 4.8 million were located in the Beijing-Tianjin-Hebei area and surrounding areas. The benefits of promoting clean fuels in the residential sector are also obvious throughout the country, and the transportation sector elevated emission standards and imposed mandatory elimination of old vehicles that do not meet emission standards [2].

## Conclusion

We analyzed the spatiotemporal changes in six air pollutants and AQI in 362 cities in China from 2015 to 2019. The national and regional air quality in China continues to improve. $PM_{2.5}$, $PM_{10}$, AQI, CO, and $SO_2$ have exhibited negative trends; $PM_{2.5}$ is the most important air pollutant in most regions in China, particularly in the Northern China, Xinjiang, Central China, and the Weihe River Basin; the spatial distribution of $NO_2$ is heterogeneous, $O_3$ and $NO_2$ pollution is an urgent problem that needs to be solved; the main reason for the change in air quality is human activities; however, the current control strategy for $PM_{2.5}$ will only par-tially reduce the $PM_{2.5}$ pollution in the Western region. Although the implementation of the "Action Plan" measures has effectively improved air quality, China's air pollution is still serious and far from the WHO standard. Measures for continuous and effective emissions control are still a top priority.

## Supporting information

**S1 File.**
(DOCX)

## Acknowledgments

I would like to express my gratitude to all those who helped me during the writing of this the-sis. My deepest gratitude goes foremost to Professor Umarova Aminat Batalbievna, my super-visor, for her constant encouragement and guidance.

## Author Contributions

**Conceptualization:** Peng Guo.

**Data curation:** Peng Guo.

**Formal analysis:** Peng Guo.

**Funding acquisition:** Peng Guo.

**Investigation:** Peng Guo, Yunqi Luan.

**Methodology:** Peng Guo.

**Project administration:** Peng Guo.

**Resources:** Peng Guo.

**Software:** Peng Guo.

**Supervision:** Peng Guo, Aminat Batalbievna Umarova.

**Validation:** Peng Guo.

**Visualization:** Peng Guo.

**Writing – original draft:** Peng Guo.

**Writing – review & editing:** Peng Guo.

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
