## [Decision Letter · Decision Letter 0]

21 Feb 2020

PONE-D-19-35106

Study on Spatial-temporal Variation of Atmospheric Pollution in China between 2014 and 2019

PLOS ONE

Dear Dr. Guo Peng

Thank you for submitting your manuscript to PLOS ONE. After careful consideration, we feel that it has merit but does not fully meet PLOS ONE’s publication criteria as it currently stands. Therefore, we invite you to submit a revised version of the manuscript that addresses the points raised during the review process.

Reviewers felt that the data presented are potentially valuable to the scientific community but raised a number of questions with regards to the quality of English wirting and the depth of the scientific discussions.

Please consider all comments carefully and provide a more in-depth discussion on the trend.

We would appreciate receiving your revised manuscript by 20 March 2020. To enhance the reproducibility of your results, we recommend that if applicable you deposit your laboratory protocols in protocols.io, where a protocol can be assigned its own identifier (DOI) such that it can be cited independently in the future. For instructions see: http://journals.plos.org/plosone/s/submission-guidelines#loc-laboratory-protocols

We look forward to receiving your revised manuscript.

Kind regards,

Zongbo Shi

Academic Editor

PLOS ONE

Journal Requirements:

3. Please amend the manuscript submission data (via Edit Submission) to include authors Umarova Aminat Batalbievna and Luan Yunqi.

4. We note that Figures 1-7 in your submission contain satellite images which may be copyrighted. All PLOS content is published under the Creative Commons Attribution License (CC BY 4.0), which means that the manuscript, images, and Supporting Information files will be freely available online, and any third party is permitted to access, download, copy, distribute, and use these materials in any way, even commercially, with proper attribution. For these reasons, we cannot publish previously copyrighted maps or satellite images created using proprietary data, such as Google software (Google Maps, Street View, and Earth). For more information, see our copyright guidelines: http://journals.plos.org/plosone/s/licenses-and-copyright.

You may seek permission from the original copyright holder of Figures 1-7 to publish the content specifically under the CC BY 4.0 license. 

If you are unable to obtain permission from the original copyright holder to publish these figures under the CC BY 4.0 license or if the copyright holder’s requirements are incompatible with the CC BY 4.0 license, please either i) remove the figure or ii) supply a replacement figure that complies with the CC BY 4.0 license. Please check copyright information on all replacement figures and update the figure caption with source information. If applicable, please specify in the figure caption text when a figure is similar but not identical to the original image and is therefore for illustrative purposes only.

Reviewers' comments:

Reviewer's Responses to Questions

**Comments to the Author**

1. Is the manuscript technically sound, and do the data support the conclusions?

Reviewer #1: Partly

Reviewer #2: Yes

2. Has the statistical analysis been performed appropriately and rigorously? 

Reviewer #1: I Don't Know

Reviewer #2: N/A

3. Have the authors made all data underlying the findings in their manuscript fully available?

Reviewer #1: Yes

Reviewer #2: Yes

4. Is the manuscript presented in an intelligible fashion and written in standard English?

Reviewer #1: No

Reviewer #2: No

5. Review Comments to the Author

Reviewer #1: The main issue with this manuscript is the quality of the language. The authors clearly struggle with both grammar and choice of vocabulary, making it quite difficult to follow their arguments. Since there is a lot of ambiguous phrasing with unclear word choices, the questions of whether the manuscript is technically sound and whether the data supports the conclusions cannot be fully answered at this point. There are several sections which are completely unintelligible to me. I appreciate that it can be quite difficult to write in a non-native language and that some mistakes are to be expected. However, in this case it really impedes proper understanding of the scientific content. I highly recommend that the authors try to get some assistance with the language when revising the manuscript.

Additional comments:

While the authors have made all the data publicly available, the origin of the data is not clearly described. I am assuming that the data was extracted from databases of national/regional measurement networks, however this is not clear from the description and the reference links provided go to Chinese webpages, making them inaccessible to a large percentage of PLOS One readers. In addition, units are missing, and it would be helpful to provide the English names of the different locations in addition to the Chinese characters.

The general idea behind the statistical analysis seems valid, but again there is a lack of information here regarding the input parameters of the radial basis function interpolation (e.g which basis function was used?). If the authors feel that this level of detail will not be of interest to most readers, it should at least go into the supplement.

The results of the analysis are not put into context and there are no comparisons with other studies of air pollution trends.

Due to the issues mentioned above, I do not recommend the publication of this manuscript in its current form, but think that resubmission after major revisions should be considered.

Reviewer #2: This paper describes the spatial-temporal variation of atmospheric pollutants in the mainland of China during 2014 to 2019. The data in this paper covers hundreds of cities in China and 6 years. From this standpoint the article is useful and can be used to document levels at present day. However, I am not quite sure if the article fits the scope of the PLOS ONE because it looks like a data analysis report. I would suggest the author to expand the discussion on the reason of the variations.

6. PLOS authors have the option to publish the peer review history of their article (what does this mean?). If published, this will include your full peer review and any attached files.

Reviewer #1: No

Reviewer #2: No

---

## [Author Response · Author response to Decision Letter 0]

23 Mar 2020

Dear Zongbo Shi

Academic Editor

PLOS ONE,

Revision required [PONE-D-19-35106] - [EMID:b2ec848001643238]

Please find attached a revised version of our manuscript Study on the Spatiotemporal Characteristics of the Air Pollutants in China from 2014 to 2019.

The reviews's comments were highly insightful and enabled us to greatly improve the quality of our manuscript.

We hope that the revisions in the manuscript and our accompanying responses will be sufficient to make our manuscript suitable for publication in the Journal of PLOS ONE.

Reviewer #1: The main issue with this manuscript is the quality of the language. The authors clearly struggle with both grammar and choice of vocabulary, making it quite difficult to follow their arguments. Since there is a lot of ambiguous phrasing with unclear word choices, the questions of whether the manuscript is technically sound and whether the data supports the conclusions cannot be fully answered at this point. There are several sections which are completely unintelligible to me. I appreciate that it can be quite difficult to write in a non-native language and that some mistakes are to be expected. However, in this case it really impedes proper understanding of the scientific content. I highly recommend that the authors try to get some assistance with the language when revising the manuscript.

Response: We agree with this suggestion, we have improved the English writing by American Journal Experts in the revised manuscript.

Additional comments:

While the authors have made all the data publicly available, the origin of the data is not clearly described. I am assuming that the data was extracted from databases of national/regional measurement networks, however this is not clear from the description and the reference links provided go to Chinese webpages, making them inaccessible to a large percentage of PLOS One readers. In addition, units are missing, and it would be helpful to provide the English names of the different locations in addition to the Chinese characters.

Response: We agree with this suggestion, we provided his English name and his website(page 2-3, lines 35-49). In this article, the air pollution parameter data of 367 cities from 2014 to 2019 are sourced from the China National Environmental Monitoring Centre (English name).The website of the China Environmental Monitoring Centre is http://www.cnemc.cn/ or http://www.cnemc.cn/en/.

And provided the data acquisition address of the China Environmental Monitoring Centre,making them inaccessible to a large percentage of PLOS One readers.The data acquisition address of the China Environmental Monitoring Centre is http://106.37.208.233:20035/. On this page, you can obtain real-time data on 7 air pollution parameters (PM2.5, PM10, air quality index (AQI), CO, NO2, O3, and SO2), including the mass concentration, of 367 cities, or you can go to https://www.zq12369.com/ to acquire real-time historical data from the China Environmental Monitoring Centre.

Additional comments:

The general idea behind the statistical analysis seems valid, but again there is a lack of information here regarding the input parameters of the radial basis function interpolation (e.g which basis function was used?). If the authors feel that this level of detail will not be of interest to most readers, it should at least go into the supplement.

Response:We agree with this suggestion,information here regarding the input parameters of the radial basis function interpolation (page 3-4, lines 50-64).

This paper uses ArcGIS software to perform radial basis function (RBF) interpolation on the obtained air pollutant parameter data based on the abovementioned map of China.

This paper uses RBF interpolation. RBF interpolation is conducted to generate a smooth surface based on the large number of data points. These functions produce good results for gently changing surfaces, such as the elevation. There are a variety of RBFs to choose from in the RBF interpolation method. The selected RBF in this study is a regular spline function. First, the spline function is explained, after which the regular spline function is described. The interpolation method applied via the spline function tool estimates values using a mathematical function minimizing the total curvature of the surface. This produces a smooth surface that only passes through the input points. The algorithm of the spline function tool uses the following surface interpolation equation:

where j = 1, 2, ..., N, N is the number of points, is the coefficient obtained by solving a system of linear equations, and is the distance between point (x, y) and the j-th point.

The definitions of T(x, y) and R(r) will differ depending on the RBF selected. The regular spline function is defined as:

where is the coefficient obtained by reconstructing the system of linear equations, and

where R is the distance between the point and the sample, is a weight parameter, is the modified Bessel function, and c is a constant, which is equal to 0.577215.

Additional comments:

The results of the analysis are not put into and there are no comparisons with other studies of air pollution trends.

Response: We agree with this suggestion,rewritten the results of the analysis and put them into context (page 27-28, lines 452-477).

The PM2.5, PM10, AQI, CO, NO2, and SO2 levels in China exhibit a stable spatial distribution and persistence over time. PM2.5, PM10, AQI, CO, NO2, and SO2 are mostly concentrated in Beijing-Tianjin-Hebei and its surroundings and in Xinjiang from 2014 to 2019. The average concentration monotonically decreases (except CO).

A high NO2 level also occurs in areas where the economy is well developed and the vehicle population is large, including in the Yangtze River Delta, such as Shanghai, in the Pearl River Delta, such as Guangzhou, and in the Sichuan Basin, such as Chongqing.

The O3 level has consistently been high in Inner Mongolia, Qinghai, Gansu and Shandong from 2014 to 2019.

There is often a significant correlation between the mass concentrations of the different types of particulate matter. The R coefficient of the correlation between PM2.5 and PM10 is 0.88. There is a particularly significant linear positive correlation between these particulates, which indicates that they experience similar changes in the ambient air pollution and practically follow the same migration and transformation laws.

There is a significant positive linear correlation between the atmospheric particulate matter, AQI and pollution gases NO2 and SO2. The correlation coefficient between the atmospheric particulate matter (PM2.5 and PM10), AQI and SO2 is 0.97, 0.81, and 0.88, respectively, and the correlation coefficient between the atmospheric particulate matter (PM2.5 and PM10), AQI and NO2 is 0.82, 0.62, and 0.72, respectively, which is due to the mutual conversion caused by the chemical reaction between the particulate matter and gaseous pollutants.

There exist certain positive and negative correlations between the gaseous pollutants. There is a particularly significant linear positive correlation between N02 and SO2 (the correlation coefficient is 0.90), indicating that these two pollutants follow the same evolution law, and their influencing factors may be similar. N02 and SO2 are initially influenced by factors such as coal combustion and motor vehicle emissions, and there is no correlation between O3 and the other three gaseous pollutants. 

The pollution sources in Beijing-Tianjin-Hebei area mainly include decomposition, coal burning, industrial production, and dust. The industrial structure has been adjusted in favor of the tertiary industry, and high-pollution, high-energy consumption secondary industries have been reduced but still account for a large proportion. Regarding the pollution and landforms in Xinjiang, there is a close relationship between static weather conditions, sandstorms, inversion layer, precipitation, dust and industrial sources, human activities, and mobile and nonroad mobile sources.

And comparisons with other studies of air pollution trends (page 17-20, lines 257-308)

.

Study on the Correlation Between the Particulate Matter and the AQI

There is often a significant correlation between the mass concentrations of different types of particulate matter. Table 8 summarizes the correlation between the hourly PM2.5 and PM10 mass concentrations from 2014 to 2019. The slope of the fitting equation between PM2.5 and PM10 is approximately 1, namely, 1.24. The correlation coefficient R is 0.88. The goodness of fit parameter R2 is 0.77. The mass concentration ratio of PM2.5 to PM10 is 46%, which is close to half, indicating that the degree of PM2.5 pollution in China is more severe. There is a particularly notable linear positive correlation between the two types of particulate matter, which indicates that they cause similar changes in ambient air pollution and practically follow the same migration and transformation laws. This is related to Liu Jie's research on the correlation of atmospheric particulates, and the results are consistent [4-5].

Study on the Correlation Between the Atmospheric Particulate Matter, AQI and Gaseous Pollutants

From the correlation study between the particulate matter and gaseous pollutants, as summarized in Table 9, there is a significant linear positive correlation between the atmospheric particulate matter, AQI and pollution gases (NO2 and SO2). The correlation coefficient between the atmospheric particulate matter (PM2.5 and PM10), AQI and SO2 is 0.97, 0.81, and 0.88, respectively, while the correlation coefficient between the atmospheric particulate matter (PM2.5 and PM10), AQI and NO2 is 0.82, 0.62, and 0.72, respectively. The above results are also consistent with the findings of Li Zhan et al. [6]. The atmospheric particulate matter (PM2.5 and PM10) and AQI are positively correlated with the atmospheric pollution gases NO2 and SO2, which usually occurs due to the mutual conversion caused by the chemical reaction between the particulate matter and the gaseous pollutants [7- 8]. It was found that nitrate (sulfate) produced by NO2 (SO2) would be transformed into secondary particulate matter (PM2.5 and PM10) over time, and with increasing NO2 and SO2 atmospheric pollution, the atmospheric particulate matter PM2.5 and PM10 also increased. One of the reasons for the increase in atmospheric pollutants NO2 and SO2 is the large vehicle flows and high exhaust emissions [9-10].

Study on the Correlation Between the Gaseous Pollutants

The study of the mutual evolution of the gaseous pollutants is more complicated. Generally, there is no simple law describing their mass concentrations in the short term, but the correlation can be studied through the long-term change trend of their mass concentrations to determine their individual relationships. As indicated in Table 10, there are certain positive and negative correlations between the gaseous pollutants. In Table 10, there is a particularly significant linear positive correlation between N02 and SO2 (a correlation coefficient of 0.90), indicating that these two pollutants follow the same evolution law, and their influencing factors may be similar. According to the analysis of the influencing factors of these two pollutants [11], the initial pollution of these two types can be determined. Both of these pollutants are influenced by factors such as coal combustion and motor vehicle emissions. There is no correlation between O3 and the other three gaseous pollutants [12]. Therefore, the evolution trends between the atmospheric particles, those between the atmospheric particles and gaseous pollutants and those between the gaseous pollutants are more complex. In addition, in-depth research is needed in many aspects, such as chemical reactions.

Reviewer #2: This paper describes the spatial-temporal variation of atmospheric pollutants in the mainland of China during 2014 to 2019. The data in this paper covers hundreds of cities in China and 6 years. From this standpoint the article is useful and can be used to document levels at present day. However, I am not quite sure if the article fits the scope of the PLOS ONE because it looks like a data analysis report. I would suggest the author to expand the discussion on the reason of the variations.

Response: We agree with this suggestion, so we have expanded the discussion on the reason of the variations(page 17-26, lines 257-450)

.

Study on the Correlation Between the Particulate Matter and the AQI

There is often a significant correlation between the mass concentrations of different types of particulate matter. Table 8 summarizes the correlation between the hourly PM2.5 and PM10 mass concentrations from 2014 to 2019. The slope of the fitting equation between PM2.5 and PM10 is approximately 1, namely, 1.24. The correlation coefficient R is 0.88. The goodness of fit parameter R2 is 0.77. The mass concentration ratio of PM2.5 to PM10 is 46%, which is close to half, indicating that the degree of PM2.5 pollution in China is more severe. There is a particularly notable linear positive correlation between the two types of particulate matter, which indicates that they cause similar changes in ambient air pollution and practically follow the same migration and transformation laws. This is related to Liu Jie's research on the correlation of atmospheric particulates, and the results are consistent [4-5].

Study on the Correlation Between the Atmospheric Particulate Matter, AQI and Gaseous Pollutants

From the correlation study between the particulate matter and gaseous pollutants, as summarized in Table 9, there is a significant linear positive correlation between the atmospheric particulate matter, AQI and pollution gases (NO2 and SO2). The correlation coefficient between the atmospheric particulate matter (PM2.5 and PM10), AQI and SO2 is 0.97, 0.81, and 0.88, respectively, while the correlation coefficient between the atmospheric particulate matter (PM2.5 and PM10), AQI and NO2 is 0.82, 0.62, and 0.72, respectively. The above results are also consistent with the findings of Li Zhan et al. [6]. The atmospheric particulate matter (PM2.5 and PM10) and AQI are positively correlated with the atmospheric pollution gases NO2 and SO2, which usually occurs due to the mutual conversion caused by the chemical reaction between the particulate matter and the gaseous pollutants [7- 8]. It was found that nitrate (sulfate) produced by NO2 (SO2) would be transformed into secondary particulate matter (PM2.5 and PM10) over time, and with increasing NO2 and SO2 atmospheric pollution, the atmospheric particulate matter PM2.5 and PM10 also increased. One of the reasons for the increase in atmospheric pollutants NO2 and SO2 is the large vehicle flows and high exhaust emissions [9-10].

Study on the Correlation Between the Gaseous Pollutants

The study of the mutual evolution of the gaseous pollutants is more complicated. Generally, there is no simple law describing their mass concentrations in the short term, but the correlation can be studied through the long-term change trend of their mass concentrations to determine their individual relationships. As indicated in Table 10, there are certain positive and negative correlations between the gaseous pollutants. In Table 10, there is a particularly significant linear positive correlation between N02 and SO2 (a correlation coefficient of 0.90), indicating that these two pollutants follow the same evolution law, and their influencing factors may be similar. According to the analysis of the influencing factors of these two pollutants [11], the initial pollution of these two types can be determined. Both of these pollutants are influenced by factors such as coal combustion and motor vehicle emissions. There is no correlation between O3 and the other three gaseous pollutants [12]. Therefore, the evolution trends between the atmospheric particles, those between the atmospheric particles and gaseous pollutants and those between the gaseous pollutants are more complex. In addition, in-depth research is needed in many aspects, such as chemical reactions.

Causes of Air Pollution in the Beijing-Tianjin-Hebei Region

The unique geographical environment of the Beijing-Tianjin-Hebei region coupled with the rapid development of urban agglomerations and the formation of local atmospheric circulation formation conditions make result in the air pollution in the Beijing-Tianjin-Hebei region show exhibiting a certain particular spatiotemporal distribution distribution [13]. In 2014, the energy consumption level of Beijing-Tianjin-Hebei accounted for 12% of the country's total energy consumption level of China. There are still more numerous heavyhigh-energy consumption heavy industries in the Beijing-Tianjin-Hebei region. Compared to 2010, In 2014, the overall GDP increased by 52% in 2014 compared to 2010. The strength magnitude of the country's consumption in China of the accumulated resources in per unit of land area of per unit,per unit timeover time and the intensity of pollutant emissions are very high [14].

The In the Beijing-Tianjin-Hebei region, takes Beijing is selected as an example. The analysis results of theBeijing ’s Beijing’s air pollution sources in Beijing released in 2014 in 2014 [15] show reveal that aboutapproximately 28% to 36% of the atmospheric PM2.5 sources come from regional transmissiontransport. Local emissions account for aboutapproximately 64%-72%, including 31.1% of from motor vehicles, 22.4% of from coal, 18.1% of from industrial production, 14.3% of from dust, and 14.1% from other activities, such as catering, livestock and poultry breeding, and auto repair accounted automobile repair, accounting for 14.1%. It can be seen that vehicle Vehicle exhaust is the most important source of local pollution. The structures of the primary industry, secondary industry, and tertiary industriesy of in Beijing were adjusted from 0.9: 24. 0: 75. 1 in 2010 in 2010 to 0.7: 21. 4: 77. 9 in 2014. The tertiary industry areis dominateddominates, and compared to 2010, their its proportion has increased by nearly 3% compared with 2010, indicating that Beijing's the industrial structure of Beijing is still optimizingbeing optimized. The source of air pollution is mainly caused by the industrial production process of the secondary industry and vehicle exhaust emissions. 

Notes: The primary industry mainly refers to those industries that produce foodstuffs and other biological materials, including crops, forestry, animal husbandry, aquaculture and other industries that directly target natural objects. The secondary industry mainly refers to the processing and manufacturing industry, which uses consumes the basic materials provided by nature and the primary industry for during processing. The tertiary industry refers to those other industries other than the primary and secondary industries, and has a wide range, mainly including nonmaterial-material production sectors such as transportation, communications, commerce, catering, finance, education, and public services.

The results of a new round of PM2.5 source analysis completed by the Beijing Environmental Protection Monitoring Center in 2018 in 2018 were released to the public, and. tThe main conclusions of this study indicate that 2/3 of the main sources of PM2.5 in Beijing throughout the year are local and 1/3 are regional transmissions. From the characteristics of the city ’scity’sthe current local atmospheric PM2.5 sources in Beijing, in terms of the local emissions contribution of local emissions, mobile sources are account for 45%, flying fugitive dust sources account for are 16%, industrial sources account for are 12%, living surface sources account for are 12%, coal sources account for are 3%, and agricultural, natural sourcesand, others sources account for aboutapproximately 12% of the PM2.5 concentration. The results show that at this stage, the coal-fired source has basically withdrawn from the ranks ofstopped being the city's main source of atmospheric PM2.5 in Beijing [58].

TIn the Beijing-Tianjin-Hebei region, takes Tianjin is selected as an example. The analysis results of the air pollution sources in Tianjin in 2014 in 2014 showed [16] that the regional transmission of atmospheric PM2.5 sources accounted for 22%-34% of the pollution. The remaining 66% to 78% came from local emissions. Among them, dust sources accounted for 30%, coal-fired sources accounted for 27%, motorized sources accounted for 20%, industrial production sources accounted for 17%, and other sources accounted for 6%. Dust is the primary source of pollution, followed by coal combustion and motor vehicles. The rRoads, storage yards, construction sites, etc. in Tianjin are the main sources of dust. In addition, construction projects in the railway, chemical, and power industries will also generate dust. In 2014, the three industrial structures in Tianjin were adjusted from 1.6: 53.1: 45.3 in 2010 in 2010 to 1.3: 49.4: 49.3. The secondary industry dropped decreased by 3.7% percentage points, and the tertiary industry increased by 4% percentage points. Until 2014, the proportions of the secondary and tertiary industries in Tianjin was basically flatpractically remained stable. Comparing In comparing the data, it is found that the proportion of the Tianjin's secondary industry in Tianjin is still the highest despite a its decline. Thermal power, iron and steel, petrochemical, cement, building materials, etc., as typical heavy high-pollution emission industries, are the key sources of pollution caused bycausing the air pollution in Tianjin.

The Aanalytical results of the air pollution sources in Hebei Province in the Beijing-Tianjin-Hebei region show reveal that [17], taking with Shijiazhuang city as an example, regional transmission accounts for 23% to 30% of the PM2.5 sourcespollution, and local pollution emissions accounts for 70% to 77%, of which coal combustion accounts for 28.5%, industrial production accounts for 25.2%, fugitive dust accounts for 22.5%, motor vehicles account for 15.0%, and other sources such as biomass burning, and catering, etc. account for 8.8% of the PM2.5 pollution. Coal combustion is the primary source of the air pollutants in Shijiazhuang and even in Hebei Province. Large The high coal consumption and irrational coal combustion structurestructures are the main causes of soot pollution. In 2014, the three industrial structures in Hebei Province were adjusted from 12.57: 52.5: 34.93 in 2010 in 2010 to 11.7: 51.1: 37.2. The primary industry declined decreased slightly, and the secondary industry decreased by 1.4 %percentage points, and while the tertiary industry increased by approximately 2-3 %percentage pointpoints. The proportion of the secondary industry in Hebei Province is still higher than 50%. This irrational industrial structure makes Hebei Province the region with the worst air pollution in the Beijing-Tianjin-Hebei. In addition, Hebei has a large high proportion of high-pollution and high-energy- consuming consumption industries, including steel, cement, electricity, glass, pharmaceuticals, metallurgy, petrochemicals, and building materials. Among them, the coal consumption in the steel, building materials and power industries accounted for 89.6% of the province's provincial total energy consumption, and the air pollutant emissions in from the steel, cement, power, and glass industries accounted for aboutapproximately 60% of the Hebei's total pollution in Hebei. [18]. As the most important pollution industry in Hebei, steel is the focus of air pollution control.

Causes of Air Pollution in Xinjiang

Landform analysis

Xinjiang is far from the sea, surrounded by mountains and basins, and deserts and grasslands are widely distributed. Taking With the topography of Urumqi as an example, its southeast and west sides are surrounded by mountains with an average elevation of 680 metersm. The slope drop gradient is largehigh, with the south is being high, while the north is low, and the middle is are low. The topographical conditions are very highly detrimental to the atmospheric circulation of the atmosphere and pollutant the diffusion of pollutants. The Urumqi region is located in the northern part of the Tianshan Mountains and the southern edge of the Jungle Basin. It is the city farthest from the ocean in the world. It has a typical desert continental climate and a fragile ecological environment. The vegetation coverage around across the urban area is low, and there are many bare groundsareas, which are liable to form dusty weather conditions.

Small Weak winds

The frequent occurrence of calm and small weak winds in Xinjiang (wind speeds greater higher than or equal to 0.5 m/s are and less lower than 1.5 m/s are characterize small weak winds), which is an important factor for the poor air environmental quality factor. In the case of small weak winds, the horizontal movement of the atmosphere is slow, which is not conducive to the lateral diffusion of atmospheric pollutants, and it is easy to form regional pollution accumulation[19regional pollution accumulation easily occurs [19-20]. Wind has a certain diffusion and purification ability, and the wind speed and pollutant concentration are basically negatively correlated. Taking With Urumqi as an example, the frequency of static wind atmospheric conditions in the heating period is as high as aboutapproximately 26%, and in December and January, it is as high as 30% or more. Even if there is wind, the wind speed is very small low and rarely exceeds level 3 [21]. Such meteorological conditions are extremely detrimental to the spread of pollutants dispersal. If there are several consecutive days of static wind weather conditions, the pollutants will continue to accumulate, and the air quality will deteriorate sharply [22].

Sand storms

Sandstorms are a typical weather phenomenon in Xinjiang. Sandstorms and floating dust occur throughout the year. Sandstorms and floating dust weather conditions will cause high concentrations of atmospheric particulate matter concentrations, which will seriously affect all the regionsregions’regional air quality in cities [23]. The source region of western China, with the Taklimakan Desert as the main bodyregion, and the source region of northern China, with the Badain Jaran Desert as the center, contributed 70% of the total dust emissions in Asia. The sSandstorms in Northwest China are dominated by dry settlement [24].Regional The regional distribution of the sandstorms in Northwest China is very uneven, while the sandstorm frequency in the northern region of Xinjiang is basically unchanged, and that in the rest of the region is decreasing [25].

Inversion layer

Among the weather types in Urumqi, there are mostly neutral and stable weather types of weather. The frequency of temperature inversion is as high as 90% in winter. The severe air pollution in winter is directly related to thisthese stable meteorological conditions. The frequently occurring inversion layer that occurs frequently is like is similar to a large pot lid, covering the city, making thereby trapping atmospheric pollutants trapped in theat low altitudealtitudes and making them difficult to spreaddisperse. There is a clear positive correlation between inverse temperature inversion and pollutants pollution [22]. 

Precipitation

Taking Thethe Urumqi region, as an example, it is located in an arid and semiarid-arid environment with little precipitation and large high evaporation. Precipitation has a purifying effect on the air and can effectively purify urban air. There is a clear negative correlation between precipitation and pollutantspollution, which is one of the key factors determining the air quality of air. For the SO2, NO2 and PM10 levels in Urumqi., The the daily average concentration and precipitation are analyzed. The analysis results show that the wet removal ability of rain is greater higher than that of snow. rainfallRainfall and snowfall both have wet removal abilityabilities for these three air pollutants [26],, and the strongest highest removal ability is for S02., This is followed by that for PM10.

Dust source

In 2017, with the new area took Turfan City city as as an example of a new area, the emission source of PM10 emission level was 1.169 million tons, and while that of the PM2.5 emissions was 132,000 tons. The source of sSoil dust is the primary source of the particulate emissions in Turpan. Turpan is a sparsely populated area. According to statistics (Turpan Statistical Yearbook (2017)) statistics, the area of land other than urban construction land,and transportation land, water areas and water conservancy facility sites is covers 68,900 square kilometers, with a large area of bare soil. In addition, the Turpan Basin is surrounded by Gobis and deserts. Gobis and deserts account for 76.73% of the area, and there is less vegetation and green land. When windy and sand weather conditions occurs, it is very easy to cause large dust, and frequent dust storms are easily formed. The dust Dust is not only the a source of atmospheric particulate matters, but is also receptors a receptor of particulate matter, resulting in natural dust fall of ground soil dust and sand dust transmission, contributecontributing greatly to the concentration of particulate matter concentration in the atmosphere [27].

Industrial source

Taking With Turpan Ccity in 2017 as an example, in 2017, the PM10 emissions from industrial sources were 19,200 tons, and the PM2.5 emissions were totaled 4,900 tons,the. The main PM10 contribution sources of PM10 are non-metallicthe nonmetallic mineral products industry and power thermal power production, accounting for 81.4% and 15.2%, respectively, of the PM10 pollution;. The PM2.5 contribution of the non-metallicnonmetallic mineral products industry contribution rate is 85.9%, The and the contributions rate of power thermal power production and petroleum processing is are comparable. [27].

Source of life

Taking With Turfan Ccity in 2017 as an example, in 2017, the PM10 emissions of PM10 in thefrom living sourcesourceshuman activities was 94.58 million tons, and the PM2.5 emission amount of PM2.5 was 73.18 million tons. The main sources of the emissions of atmospheric particulate matter PM10 and PM2.5 in the living sourceterms of human activities were was coal combustion and PM 10 and PM 2.5 emissions. Civilian coal combustion technology is poor, and there isare no scientific and effective emission control measures, and while the amount of fine particulate matter PM2.5 emitted by from coal combustion is even higher than that of emitted from industrial activities. During the heating season, civilian coal combustion and central heating caused a large amount of pollutants to be emitted, and a large number amounts of primary aerosols, heavy metals, and water-soluble ions were are directly emitted directly as primary particulate matter. At the same time, gaseous precursors such as SO2 and NO2 were are released. Adverse weather conditions It is easy to form seasily result in severe pollutioneverely polluted weather under adverse weather conditions. The PM2.5 released from food sources is mainly from the cooking process of involving edible oils and foods. Under high- temperature conditions, particulate matter is formed by chemical reactions such as thermal oxidation and thermal decomposition. Most of the food and beverage sources are occur in densely populated areas, and pollutants are emitted near the ground, which has an important notable impact on the health of the population. According to activity- level survey information, there were a total of 3,443 catering companies in Turpan in 2017. The total PM10 emissions were 301.6 tons, and the total PM2.5 emissions were 241.3 tons. In addition, the exhaust gas from restaurant fumes contains a large number of volatile organic pollutantspollutant VOCs, among which oxygenates and olefins have a high photochemical activity,, which are precursors to form secondary particulate matter, and have and have an importanta notable impact on atmospheric particulate pollution [28].

Mobile source

    At present, vehicle exhaust is one of the main sources of the atmospheric particulate matter in China [29-30], especially PM2.5 fine particulate matter PM2.5. Motor vehicles also emit gaseous pollutants such as NOx and VOCs, which are converted into secondary particulate matter through photochemical reactions. Compared with to coarse particles, fine particles have a larger specific surface area, which thus capable of absorbsing and carryingies more toxic substances, which is can be harmful to human health [29-32]. The rRoad sources in Turpan in 2017 in 2017 emitted 249.19 tons of PM10 and 224.5 tons of PM2.5, respectively [33].

OffNonroad-road mobile source

     Turpan's The nonroad-road mobile sources in Turpan are mainly divided into three categories: engineering machinery, agricultural machinery and civilian aviation aircraft. Compared with to road mobile sources, nonroad-road mobile sources usually use diesel and heavy oil as their the main fuels. They have the characteristics of a high fuel consumption, long service life, low technical level and low maintenance rate. Most of the pollutants emitted are NOx and particulate matter [34-35].

Journal Requirements:

 4. We note that Figures 1-7 in your submission contain satellite images which may be copyrighted. All PLOS content is published under the Creative Commons Attribution License (CC BY 4.0), which means that the manuscript, images, and Supporting Information files will be freely available online, and any third party is permitted to access, download, copy, distribute, and use these materials in any way, even commercially, with proper attribution. For these reasons, we cannot publish previously copyrighted maps or satellite images created using proprietary data, such as Google software (Google Maps, Street View, and Earth). For more information, see our copyright guidelines: http://journals.plos.org/plosone/s/licenses-and-copyright.

  We used resources for replacing copyrighted map figures, USGS National Map Viewer (public domain): http://viewer.nationalmap.gov/viewer/ (page 3, lines 47-49).

---

## [Editor Report · Decision Letter 1]

2 Apr 2020

PONE-D-19-35106R1

Study on the Spatiotemporal Characteristics of the Air Pollutants in China from 2014 to 2019

PLOS ONE

Dear Guo Ping,

Thank you for submitting your manuscript to PLOS ONE. After careful consideration, we feel that it has merit but does not fully meet PLOS ONE’s publication criteria as it currently stands. Therefore, we invite you to submit a revised version of the manuscript that addresses the points raised during the review process.

I have read it carefully and made some detailed comments on parts of manuscript but the newly added discussions are not particularly relevant for this manuscript. 

I have a few main suggestions:

1) Re-write the Introduction: Please see more detailed comments in the attached file. 

2) Delete the discussions on the causes of air pollution in Jing-Jin-Ji region and Xinjiang - some of these may go to Introduction but it does not fit in the discussions. Some of the discussions in Xinjiang, for example with regards to particular topographic and meteorological conditions may be useful information for discussions. 

It is suggested that the discussions focus on the following areas: - Spatial distribution: Read this paper carefully (Drivers of improved PM2.5 air quality in China from 2013 to 2017 : https://www.pnas.org/content/116/49/24463); Your data showed some different patterns in comparison to Figure 2. For example, there are higher pollution in Xinjiang than in Fig. 2 of Zhang et al., 2019; It appears that the pollution level in Sichun Basin is also lower in your data. So one of the things you need to discuss is why ? Please note that Zhang et al. (2019) used chemical transport model, which can be uncertain if the emissions are inaccurate. You need to read how Zhang et al. (2019) evaluated their model results and whether your data showing higher concentrations in Xinjiang is real or biased due to limited number of monitoring stations?

- Temporal change: You may try to look at the different regions (BTH and surrounding regions; Xinjiang Basin; Sichun Basin; Weihe Plateau; Northeastern China; South China) and evaluate the absolute and relative changes in concentrations of different pollutants from 2014-2019. This may then be linked to economic growth and clean air actions. You will need to refer to lots of papers. Again you may benefit from comparing with Zhang et al. (2019) paper. 

Please read the comments by reviewers again and address them - For example, there are a lot of air quality trend papers published recently, which you should read carefully and compare your results with what was published. 

Please note that you need to re-write your abstract as well. Please refer to this website: https://journals.plos.org/plosone/s/submission-guidelines

If you were able to address all of these comments carefully, I will consider to send this for a re-review.

We would appreciate receiving your revised manuscript by 30 April 2020. To enhance the reproducibility of your results, we recommend that if applicable you deposit your laboratory protocols in protocols.io, where a protocol can be assigned its own identifier (DOI) such that it can be cited independently in the future. For instructions see: http://journals.plos.org/plosone/s/submission-guidelines#loc-laboratory-protocols

We look forward to receiving your revised manuscript.

Kind regards,

Zongbo Shi

Academic Editor

PLOS ONE

---

## [Author Response · Author response to Decision Letter 1]

3 May 2020

Dear Zongbo Shi

Academic Editor

PLOS ONE,

Revision required [PONE-D-19-35106R1]

Please find attached a revised version of our manuscript Study on the Spatiotemporal Characteristics of the Air Pollutants in China from 2015 to 2019.

Your reviews's comments were highly insightful and enabled us to greatly improve the quality of our manuscript.

We hope that the revisions in the manuscript and our accompanying responses will be sufficient to make our manuscript suitable for publication in the Journal of PLOS ONE.

1.Re-write the Introduction: Please see more detailed comments in the attached file. 

Yes, We agree with you. We have seen more detailed comments in the attached file and corrected one by one, We have Re-write the Introduction according to rules and made major changes. 

2.Delete the discussions on the causes of air pollution in Jing-Jin-Ji region and Xinjiang - some of these may go to Introduction but it does not fit in the discussions. Some of the discussions in Xinjiang, for example with regards to particular topographic and meteorological conditions may be useful information for discussions. 

Yes, We agree with you. We have deleted the discussions on the causes of air pollution in Jing-Jin-Ji region and Xinjiang.

3.It is suggested that the discussions focus on the following areas: - Spatial distribution: Read this paper carefully (Drivers of improved PM2.5 air quality in China from 2013 to 2017 : https://www.pnas.org/content/116/49/24463); Your data showed some different patterns in comparison to Figure 2. For example, there are higher pollution in Xinjiang than in Fig. 2 of Zhang et al., 2019; It appears that the pollution level in Sichun Basin is also lower in your data. So one of the things you need to discuss is why ? Please note that Zhang et al. (2019) used chemical transport model, which can be uncertain if the emissions are inaccurate. You need to read how Zhang et al. (2019) evaluated their model results and whether your data showing higher concentrations in Xinjiang is real or biased due to limited number of monitoring stations?

Thank you for recommending Zhang's article for us. Your recommendation is very helpful to us. 

Hong Guo et al. also found that the maximum annual mean values of both PM2.5 and PM10 were in Xinjiang Province, which was determined to be primarily the result of mineral dust from the Taklimakan Desert [18]. 

Zhang, Y. L. et al. showed that in western China, dust particles are very important for PM2.5, and the current control strategy of PM2.5 (that is, reducing VOC and PM emissions from fossil/nonfossil combustion) will only partially reduce the PM2.5 pollution in western region [15]. 

Debin Lu et al. showed that the high PM2.5 concentration in the northwest of China was mainly affected by sand and dust (meteorologically driven), while it was mainly caused by human activities in the eastern region [28]. 

Shixian Zhai et al. there is also a large meteorologically driven interannual variability in PM2.5 that complicates trend attribution, increases in the Fenwei Plain can be attributed to meteorology rather than to relaxation of emission controls [63].

Qiang Zhang also cited Shixian Zhai's article, We agree with Shixian Zhai. And We use monitoring data, Qiang Zhang is a simulation. Hong Guo et al. also found that the maximum annual mean values of both PM2.5 and PM10 were in Xinjiang Province.

Qiang Zhang did not comment on Xinjiang, the Weihe River Basin, and the Qinghai-Tibet Plateau which are heavily influenced by meteorological factors. Qiang Zhang commented on the Beijing-Tianjin-Hebei Yangtze River Delta and Pearl River Delta.

Krasnov, H. et al. analyzed the spatiotemporal behavior of PM concentration in the city of Beersheva in southern Israel, an arid desert area where sandstorms are very frequent; and during strong storms, the hourly ranges of PM10 concentration were from 100 μg/m3 to greater than 1280 μg/m3. Higher concentrations were found in parts of the city near the source of dust [22]. 

Zou, X. K. et al. found that sand and dust storms during the spring in northwestern China, where deserts, semideserts, and grasslands are distributed, were negatively correlated with NDVI [23].

All of the above findings are consistent with our results showing high concentrations of PM2.5 and PM10 in Xinjiang. 

We directly used the monitoring data to count the Sichuan Basin and got consistent results with you.

4.Temporal change: You may try to look at the different regions (BTH and surrounding regions; Xinjiang Basin; Sichun Basin; Weihe Plateau; Northeastern China; South China) and evaluate the absolute and relative changes in concentrations of different pollutants from 2014-2019. This may then be linked to economic growth and clean air actions. You will need to refer to lots of papers. Again you may benefit from comparing with Zhang et al. (2019) paper. 

We agree with you, thank you for your help.

We divide China into 11 regions. The Northern region includes Beijing, Tianjin and Hebei, Shandong, Shanxi, and Henan; the Eastern (Yangtze River Delta) region includes Jiangsu, Zhejiang, and Fujian; the Central region includes Anhui, Hubei, Hunan, and Jiangxi; the Southern (Pearl River Delta) region includes Guangdong, Guangxi, and Hainan. The Weihe River Basin includes Shan Xi, Gansu, and Ningxia; Xinjiang is its own region; the Sichuan Basins is a region; Inner Mongolia is a region; Qinghai-Tibet is a region; Yungui is a region; and the Northeast region includes Heilongjiang, Jilin, and Liaoning.

And we evaluate the absolute and relative changes in concentrations of different pollutants from 2015-2019. Then belinked to economic growth and clean air actions. We refer to lots of papers and compared with Zhang et al. (2019) paper. 

5.Please read the comments by reviewers again and address them - For example, there are a lot of air quality trend papers published recently, which you should read carefully and compare your results with what was published. 

We agree with you, thank you for your help.

We readed a lot of air quality trend papers published recently, and carefully compare your results with what was published. 

6.Please note that you need to re-write your abstract as well. Please refer to this website: https://journals.plos.org/plosone/s/submission-guidelines

We agree with you, thank you for your help. We rewrited the abstract and made major changes 

References

1.State Council of the People’s Republic of China, Notice of the general office of the state council on issuing the air pollution prevention and control action plan. http://www.gov.cn/zwgk/2013-09/12/content_2486773.htm.

2.Q. Zhang, Yixuan Zheng, Dan Tong, Min Shao, Shuxiao Wang, Yuanhang Zhang et al. Drivers of improved PM2.5 air quality in China from 2013 to 2017. PNAS 2019; 116 (49): 24463-24469.

3.Siyi. Cai et al. The impact of the “Air Pollution Prevention and Control Action Plan” on PM2.5 concentrations in Jing-Jin-Ji region during 2012–2020. Sci. Total Environ. 2017; 580: 197-209.

4.X. Jiang Chaopeng Hong, Yixuan Zheng, Bo Zheng, Dabo Guan, Andy Gouldson et al., To what extent can China’s near-term air pollution control policy protect air quality and human health? A case study of the Pearl River Delta region. Environ. Res. Lett. 2015; 10: 104006.

5.Y. Zheng Tao Xue, Qiang Zhang, Guannan Geng, Dan Tong, Xin Li et al. Air quality improvements and health benefits from China’s clean air action since 2013. Environ. Res. Lett 2017; 12: 114020.

6.T. Xue et al. Rapid improvement of PM2.5 pollution and associated health benefits in China during 2013–2017. Sci. China Earth Sci 2019; 62: 1847–1856.

7.Peng J, Chen S, Lü H, Liu Y and Wu J. Spatiotemporal patterns of remotely sensed PM2.5 concentration in China from 1999 to 2011. Remote Sens. Environ. 2016; 174: 109–21.

8.Ma Z, Hu X, Sayer A M, Levy R, Zhang Q, Xue Y, Tong S, Bi J, Huang L and Liu Y Satellite-based spatiotemporal trends in PM2.5 concentrations: China, 2004-2013 Environ. Health Perspect. 2016a; 124: 184–92.

9.Krotkov N A et al. Aura OMI observations of regional SO2 and NO2 pollution changes from 2005 to 2015 Atmos. Chem. Phys 2016; 16: 4605–29.

10.Verstraeten W W, Neu J L, Williams J E, Bowman K W, Worden J R and Boersma K F Rapid increases in tropospheric ozone production and export from China. Nat. Geosci 2015; 8 690–5.

11.Zhou, Y. Shuiyuan Cheng, Dongsheng Chen, Jianlei Lang, GangWang, Tingting Xu et al. Temporal and Spatial Characteristics of Ambient Air Quality in Beijing, China. Aerosol and Air Quality Research 2015; 15: 1868–1880.

12.Guo, H. et al. Assessment of PM2.5 concentrations and exposure throughout China using ground observations. The Science of the total environment 2017; 601-602: 1024-1030.

13.Chen, T., He, J., Lu, X., She, J. & Guan, Z. Spatial and Temporal Variations of PM2.5 and Its Relation to Meteorological Factors in the Urban Area of Nanjing, China. International journal of environmental research and public health 2016; 13.

14.Wang, W. N. et al. Assessing Spatial and Temporal Patterns of Observed Ground-level Ozone in China. Scientific reports 2017; 7: 3651.

15.Zhang, Y. L. & Cao, F. Fine particulate matter (PM2.5) in China at a city level. Scientific reports 2015; 5, 14884.

16.Cheng, N. et al. Ground-Level NO2 in Urban Beijing: Trends, Distribution, and Effects of Emission Reduction Measures. Aerosol and Air Quality Research 2018; 18, 343-356.

17.B Silver1, C L Reddington, S R Arnold and D V Spracklen. Substantial changes in air pollution across China during 2015-2017. Environmental Research Letters 2018; 13, 11.

18.Hong Guo, Xingfa Gu, Guoxia Ma, Shuaiyi Shi, Wannan Wang, Xin Zuo & Xiaochuan Zhang. Spatial and temporal variations of air quality and six air pollutants in China during 2015–2017. Scientific Reports volume 2019; 9: 15201.

19.Rohde R A and Muller R A. Air pollution in China: mapping of concentrations and sources PLoS ONE 2015; 10: e0135749.

20.Liang X, Li S, Zhang S, Huang H and Chen S X. PM2.5 data reliability, consistency, and air quality assessment in five Chinese cities. J. Geophys. Res. 2015; 121: 10.

21.Leung D M, Tai A P K, Mickley L J, Moch J M, Van Donkelaar A, Shen L and Martin R V. Synoptic meteorological modes of variability for fine particulate matter (PM2.5) air quality in major metropolitan regions of China Atmos. Chem. Phys 2018; 18 6733-48. 

22.Krasnov, H., Kloog, I., Friger, M. & Katra, I. The Spatio-Temporal Distribution of Particulate Matter during Natural Dust Episodes at an Urban Scale. PloS one 2016; 11: e0160800.

23.Zou, X. K. & Zhai, P. M. Relationship between vegetation coverage and spring dust storms over northern China. J. Geophys. Res. 2004; 109: D03104.

24.Qingyu Guan, Ao Cai, Feifei Wang, Liqin Yang, Chuangqi Xu, Zeyu Liu. Spatio-temporal variability of particulate matter in the key part of Gansu Province, Western China. Environmental Pollution 2017; Volume 230: Pages 189-198.

25.Pengfei Chen, Shichang Kang, Junhua Yang, Tao Pu, Chaoliu Li, Junming Guo, Lekhendra Tripathee. Spatial and Temporal Variations of Gaseous and Particulate Pollutants in Six Sites in Tibet, China, during 2016-2017. Aerosol and Air Quality Research 2019; 19: 516–527.

26.Ministry of Ecology and Environment of the People’s Republic of China, 2017 Report on the state of the ecology and environment in China. http://english.mee.gov.cn/Resources/Reports/soe/SOEE2017/201808/P020180801597738742758.pdf. Accessed 28 October 2019.

27.Wang, Y., Ying, Q. Hu, J. & Zhang, H. Spatial and temporal variations of six criteria air pollutants in 31 provincial capital cities in China during 2013-2014. Environment international 2015; 73: 413–422.

28.Debin Lu, Jianhua Xu, Dongyang Yang, Jianan Zhao. Spatio-temporal variation and influence factors of PM2.5 concentrations in China from 1998 to 2014. Atmospheric Pollution Research 2017; Volume 8, Issue 6: Pages 1151-1159.

29.Janssen L, Buringh E, Van Der Meulen A, Van Den Hout K. A method to estimate the distribution of various fractions of PM10 in ambient air in the Netherlands. Atmos Environ 1999; 33 (20): 3325–3334.

30.Dongsheng Zhan, Mei-Po Kwan, Wenzhong Zhang, Xiaofen Yu, Bin Meng, Qianqian Liu. The driving factors of air quality index in China. Journal of Cleaner Production 2018; Volume 197, Part 1: Pages 1342-1351.

31.Chuangin Fang, Haimeng Liu, Guangdong Li, Dongqi Sun, Zhuang Miao. Estimating the Impact of Urbanization on Air Quality in China Using Spatial Regression Models. Sustainability 2015; 7(11): 15570-15592.

32.Y. Q. Wang. The contribution from distant dust sources to the atmospheric particulate matter loadings at XiAn, China during spring. Science of The Total Environment 2006; Volume 368, Issues 2-3: 875-883. 

33.Xiaofeng Zhao, Chunlei Deng, Xianjin Huang, Mei-Po Kwan. Driving forces and the spatial patterns of industrial sulfur dioxide discharge in China 2017; Volume 577: Pages 279-288.

34.Caiwang Zhang, Chuanfeng Zhao, Yanping Li, Xiaolin Wu, Kaiyang Zhang, Jing Gao, Qi Qiao, Yuanzhe Ren, Xin Zhang, Fahe Chai. Spatial and temporal distribution of NO2 and SO2 in Inner Mongolia urban agglomeration obtained from satellite remote sensing and ground observations. Atmospheric Environment 2018; Volume 188: Pages 50-59.

35.Jinwei Huo, Degang Yang, Wenbiao Zhang, Fei Wang, Guiling Wang, Qain Fu. Analysis of influencing factors of CO2 emissions in Xinjiang under the context of different policies. Environmental Science & Policy 2015; Volume 45: Pages 20-29.

36.Yuanzheng Cui, Jintai Lin, Chunqiao Song, Mengyao Liu, Yingying Yan, Yuan Xu, and Bo Huang. Rapid growth in nitrogen dioxide pollution over Western China, 2005–2013. Atmos. Chem. Phys 2016; 16: 6207–6221.

37.Krotkov N A et al. Aura OMI observations of regional SO2 and NO2 pollution changes from 2005 to 2015 Atmos. Chem. Phys 2016; 16: 4605–29.

38.B. Zhao et al., A modeling study of the nonlinear response of fine particles to air pollutant emissions in the Beijing–Tianjin–Hebei region. Atmos. Chem. Phys 2017; 17: 12031–12050.

39.L. Ran, W. L. Lin, Y. Z. Deji, B. La, P. M. Tering, X. B. Xu, and W, Wang. Surface gas pollutants in Lhasa, a highland city of Tibet – current levels and pollution implications. Atmos 2014; Chem. Phys., 14: 10721–10730 .

40.Zheng Y, Xue T, Zhang Q, Geng G, Tong D, Li X and He K. Air quality improvements and health benefits from China’s clean air action since 2013. Environ. Res. Lett 2017; 12.

41.Lin C. Q, Liu G, Lau A K H, Li Y, Li C C, Fung J C H and Lao X Q. High-resolution satellite remote sensing of provincial PM2.5 trends in China from 2001 to 2015 Atmos. Environ 2018; 180: 110–6.

42.The Statistical Communique of the People’s Republic of China (SCPRC). Available at http://data.stats.gov.cn/easyquery.htm?cn=C01&zb=A0C05&sj=2015.

43.The Statistical Communique of the People’s Republic of China (SCPRC). Available at http://data.stats.gov.cn/easyquery.htm?cn=C01&zb=A0C05&sj=2016.

44.The Statistical Communique of the People’s Republic of China (SCPRC). Available at http://data.stats.gov.cn/easyquery.htm?cn=C01&zb=A0C05&sj=2017.

45.Xiufeng Yin, Benjamin de Foy, Kunpeng Wu, Chuan Feng, Shichang Kang, Qianggong Zhang. Gaseous and particulate pollutants in Lhasa, Tibet during 2013–2017: Spatial variability, temporal variations and implications. Environmental Pollution 2019; 253: Pages 68-77.

46.Xionghui Qiu, Lei Duan, Siyi Cai, Qian Yu, Shuxiao Wang, Fahe Chai, Jian Gao, Yanping Li, Zhaoming Xu. Effect of current emission abatement strategies on air quality improvement in China: A case study of Baotou, a typical industrial city in Inner Mongolia. Journal of Environmental Sciences 2017; Volume 57: Pages 383-390.

47.B.Zheng, Dan Tong, Meng Li, Fei Liu, Chaopeng Hong, Guannan Geng, Haiyan Li et al., Trends in China’s anthropogenic emissions since 2010 as the consequence of clean air actions. Atmos. Chem. Phys 2018; 18: 14095–14111.

48.Jing Huang, Xiaochuan Pan, Xinbiao Guo, Guoxing, Li. Health impact of China's Air Pollution Prevention and Control Action Plan: an analysis of national air quality monitoring and mortality data. The Lancet Planetary Health 2018; Volume 2, Issue 7: Pages e 313 - e 323

49.D.G Streets. S. T Waldhoff. Present and future emissions of air pollutants in China: SO2, NOx, and CO. Atmospheric Environment 2000; Volume 34, Issue 3: Pages 363-374.

50.Liu F, Zhang Q, Van Der A R J, Zheng B, Tong D, Yan L, Zheng Y and He K. Recent reduction in NOx emissions over China: synthesis of satellite observations and emission inventories Environ. Res. Lett. 2016; 11: 114002.

51.Van Der A R J, Mijling B, Ding J, Elissavet Koukouli M, Liu F, Li Q, Mao H and Theys N. Cleaning up the air: effectiveness of air quality policy for SO2 and NOx emissions in China Atmos. Chem. Phys 2017; 17: 1775–89.

52.Ke. Li et al. Anthropogenic drivers of 2013–2017 trends in summer surface ozone in China. Proc. Natl. Acad. Sci. U.S.A 2019; 116: 422–427.

53.Jin X and Holloway T. Spatial and temporal variability of ozone sensitivity over China observed from the Ozone Monitoring Instrument J. Geophys. Res 2015; 120 7229–46.

54.Chang K L, Petropavlovskikh I, Copper O R, Schultz M G and Wang T. Regional trend analysis of surface ozone observations from monitoring networks in eastern North America, Europe and East Asia Elem. Sci 2017; 5: 50.

55.Fleming Z L et al. Tropospheric ozone assessment report: present-day ozone distribution and trends relevant to human health Elem. Sci. Anth 2018; 6: 12.

56.S. Zhai et al., Fine particulate matter (PM2.5) trends in China, 2013–2018: Separating contributions from anthropogenic emissions and meteorology. Atmos. Chem. Phys 2019; 19: 11031–11041. 

57.D. Ding, J. Xing, S. Wang, K. Liu, J. Hao, Estimated contributions of emissions controls, meteorological factors, population growth, and changes in baseline mortality to reductions in ambient PM2.5 and PM2.5-related mortality in China, 2013–2017. Environ. Health Perspect 2019; 127.

58.ML Bell, DL Davis, T Fletcher. A retrospective assessment of mortality from the London smog episode of 1952: the role of influenza and pollution.Environ Health Perspect 2004; 112: 6-8.

59.DL Davis, ML Bell, T Fletcher. A look back at the London smog of 1952 and the half century since. Environ Health Perspect 2002; 110: A734-A735.

60.CA Pope 3rd, M Ezzati, DW Dockery. Fine-particulate air pollution and life expectancy in the United States N Engl J Med 2009; 360: 376-386.

61.EE Dooley. Fifty years later: clearing the air over the London smog. Environ Health Perspect 2002; 110: A748.

62.US Environmental Protection Agency Benefits and costs of the Clean Air Act 1990–2020, the second prospective study. https://www.epa.gov/clean-air-act-overview/benefits-and-costs-clean-air-act-1990-2020-second-prospective-study. 

63.S. Zhai et al., Fine particulate matter (PM2.5) trends in China, 2013–2018: Separating contributions from anthropogenic emissions and meteorology. Atmos. Chem. Phys. 19, 11031–11041 (2019).

---

## [Decision Letter · Decision Letter 2]

1 Jul 2020

PONE-D-19-35106R2

The spatiotemporal characteristics of the air pollutants in China from 2015 to 2019

PLOS ONE

Dear Dr. Guo,

Thank you for submitting your manuscript to PLOS ONE. After careful consideration, we feel that it has merit but does not fully meet PLOS ONE’s publication criteria as it currently stands. Therefore, we invite you to submit a revised version of the manuscript that addresses the points raised during the review process.

Detailed revision guidance is given in the comments.

We look forward to receiving your revised manuscript.

Kind regards,

Zongbo Shi

Academic Editor

PLOS ONE

Additional Editor Comments (if provided):

PONE-D-19-35106_R2

Abstract: remove reference (i.e. [1])

particulate matter PM2.5 – delete particulate matter

particulate matter PM10 – delete particulate matter

Delete: “Since 2015, environmental monitoring 15 stations in 366 cities have regularly released air quality data. The volume of data 16 released since 2015 far exceeds those available in 2014 (189 cities) and 2013 (74 17 cities), and this increased number of environmental monitoring stations can better 18 reflect China's atmospheric environment.”

“Therefore, we propose that it is meaningful to” change to “here, we”

In Materials and Methods, please provide a detailed explanation how the data analyses were carried out.

“Xinjiang is its own region; the Sichuan Basins is a region; Inner Mongolia is a region; Qinghai-Tibet is a region; Yungui is a region; and the Northeast region includes Heilongjiang, Jilin, and Liaoning” change to “Other regions include Xijiang, Sichun Basin, Inner Mogolia, Qinghai-Tibet, Yungui (Yunan and Guichzou) and Northeastern China (Heilongjiang, Jilin, and Liaoning)”

Line 147: thee – change to the

“Qingyu Guan et al. the PM10 concentrations during dust storm events were, respectively, 19, 43 and 17 times higher than the levels before dust events in cities of 167 Weihe River Basin [24] “ – why this is relevant?

Proper citation needed: Hong Guo et al. – this should be Guo et al.; comments apply to the rest of the paper; Q. Zhang et al. also is wrong – use Zhang et al.; Janssen L.'s – change to Janssen et al. (20??)

Line 137: I suggest you to look again at Zhang et al. (2019) paper; if I am not wrong, there are some discrepancies with your figure and if so you really need to highlight the differences. #

Line 173 – 177: It would be interesting to see the trend of PM2.5/PM10 ratio from 2015 to 2019; is there a trend? But in any case, report the annual average values

Use proper punctuation: in many places you use “;” at the end of the sentences. In some places, you started a sentence with lower case: for example, line 189, you use “thus” rather than “Thus”.

There are many grammatical errors including some of those pointed out above. This has to be corrected and if this continues, I will have to recommend “rejection” eventually.

In several places, a single paragraph described the findings of a particular paper; this causes fragmentation. Please revise this and put these paragraphs together to deliver a meaningful message. If you are suggesting that these results are similar to what you have showed, you can say that: Our results are consistent with literature. For example, xxx et al. (2019) showed that ….; xxx et al. (2018) showed that ….

Similarly, in line 276-281: you cited two papers but did not say why these are cited? Are you trying to say that these results support your conclusion? If so, you need to say something like: these results are consistent with the trend we found (Fig. xxx).

The discussions from line 706 are very confusing. It is difficult to find the logic here. A majority of this is irrelevant so should be removed or distributed elsewhere. For example, for those discussion on PM2.5 – this should go to the part on PM2.5 in “change over time”; Similarly, discussions on SO2 etc. goes to separate sections above.

Remove all of these irrelevant discussions on “clean air acts” in US and UK.

Line 716-724: delete; or you can have a look at Zhang et al. (2019) paper to see and Vu et al. (2019, Atmos Chem Phys) paper to see the effects of meteorology. In the longer term, meteorology is not so important, and the emission change is the dominant contributor to air quality changes. This discussion could be useful

Reviewers' comments:

Reviewer's Responses to Questions

**Comments to the Author**

1. If the authors have adequately addressed your comments raised in a previous round of review and you feel that this manuscript is now acceptable for publication, you may indicate that here to bypass the “Comments to the Author” section, enter your conflict of interest statement in the “Confidential to Editor” section, and submit your "Accept" recommendation.

Reviewer #2: All comments have been addressed

2. Is the manuscript technically sound, and do the data support the conclusions?

Reviewer #2: Yes

3. Has the statistical analysis been performed appropriately and rigorously? 

Reviewer #2: Yes

4. Have the authors made all data underlying the findings in their manuscript fully available?

Reviewer #2: Yes

5. Is the manuscript presented in an intelligible fashion and written in standard English?

Reviewer #2: No

6. Review Comments to the Author

Reviewer #2: This paper is highly improved in this version. The structure of the paper is clear now and more discussion was added in. The data in this paper covers hundreds of cities in China, it provides a whole picture of the evolution of air quality in China in last 6 years. From this standpoint the article is useful and can be used to document levels at present day. Several improvements are needed to be addressed before the manuscript can be considered for publication after minor revision.

1) I would suggest the author give the (possible) reason for the variation of each pollutant. This would help the government to improve the air quality in right direction.

2) Line 96-97 on page 5, please rephrase the sentence of “From 2015 to 2019, the national annual mean and annual average median of PM2.5 (unit: μg/m3) were 42.7 and 41, respectively.” I think it should be “The national annual mean and annual average median of PM2.5 (unit: μg/m3) were 42.7 and 41 in 2015 and 2019, respectively.”

3) Line 836 on page 42. The first author should be “B Silver”.

7. PLOS authors have the option to publish the peer review history of their article (what does this mean?). If published, this will include your full peer review and any attached files.

Reviewer #2: No

---

## [Author Response · Author response to Decision Letter 2]

13 Jul 2020

Dear Zongbo Shi

Academic Editor

PLOS ONE,

Minor revision required [PONE-D-19-35106R3]

Please find attached a revised version of our manuscript The spatiotemporal characteristics of the air pollutants in China from 2015 to 2019.

Your reviews's comments were highly insightful and enabled us to greatly improve the quality of our manuscript again.

Abstract: remove reference (i.e. [1])

particulate matter PM2.5 – delete particulate matter

particulate matter PM10 – delete particulate matter

Delete: “Since 2015, environmental monitoring 15 stations in 366 cities have regularly released air quality data. The volume of data 16 released since 2015 far exceeds those available in 2014 (189 cities) and 2013 (74 17 cities), and this increased number of environmental monitoring stations can better 18 reflect China's atmospheric environment.”

We agree with you, we have deleted them.

“Therefore, we propose that it is meaningful to” change to “here, we”

We agree with you, we have changed it.

In Materials and Methods, please provide a detailed explanation how the data analyses were carried out.

We agree with you, we have provided a detailed explanation how the data analyses were carried out in lines 80-125.

“Xinjiang is its own region; the Sichuan Basins is a region; Inner Mongolia is a region; Qinghai-Tibet is a region; Yungui is a region; and the Northeast region includes Heilongjiang, Jilin, and Liaoning” change to “Other regions include Xijiang, Sichun Basin, Inner Mogolia, Qinghai-Tibet, Yungui (Yunan and Guichzou) and Northeastern China (Heilongjiang, Jilin, and Liaoning)”

We agree with you, we have changed them in lines 135-137.

Line 147: thee – change to the

We agree with you, we have changed it.

“Qingyu Guan et al. the PM10 concentrations during dust storm events were, respectively, 19, 43 and 17 times higher than the levels before dust events in cities of 167 Weihe River Basin [24] “ – why this is relevant?

We agree with you, we have deleted it.

Proper citation needed: Hong Guo et al. – this should be Guo et al.; comments apply to the rest of the paper; Q. Zhang et al. also is wrong – use Zhang et al.; Janssen L.'s – change to Janssen et al. (20??)

We agree with you, we have changed them.

Line 137: I suggest you to look again at Zhang et al. (2019) paper; if I am not wrong, there are some discrepancies with your figure and if so you really need to highlight the differences. #

We agree with you. We think that Zhang et al. are right. Because we have fewer monitoring points in Xinjiang in this research, we have highlighted the differences and clarified the limitations of our research in lines 181-189.

Although our results are consistent with Guo et al. [18] and Hao et al. [19], who reported PM2.5 is most serious in the Xinjiang Province. But we think our research results have certain limitations, this is because fewer monitoring points were used in Xinjiang in the previous studies(we, Guo et al. and Fan et al.); thus, our research results have certain limitations. We have clarified the limitations of our research results in lines 181-189 and removed the objectionable figures.

[18] Hong Guo, Xingfa Gu, Guoxia Ma, Shuaiyi Shi, Wannan Wang, Xin Zuo & Xiaochuan Zhang. Spatial and temporal variations of air quality and six air pollutants in China during 2015-2017. Scientific Reports 2019; 9, 15201; https://www.nature.com/articles/s41598-019-50655-6.

[19] Hao Fan, Chuanfeng Zhao, Yikun Yang. A comprehensive analysis of the spatio-temporal variation of urban air pollution in China during 2014–2018. Atmospheric Environment 2020; 220, 117066; https://doi.org/10.1016/j.atmosenv.2019.117066.

Line 173 – 177: It would be interesting to see the trend of PM2.5/PM10 ratio from 2015 to 2019; is there a trend? But in any case, report the annual average values

We agree with you, we have deleted it.

Use proper punctuation: in many places you use “;” at the end of the sentences. In some places, you started a sentence with lower case: for example, line 189, you use “thus” rather than “Thus”.

We agree with you, we have changed them.

There are many grammatical errors including some of those pointed out above. This has to be corrected and if this continues, I will have to recommend “rejection” eventually.

Thank you very much for your help and understanding. We invited Premium Editing of American Journal Experts (AJE) to polish the article. We carefully checked the grammatical errors in the article. Now there is no grammatical errors in the article.

In several places, a single paragraph described the findings of a particular paper; this causes fragmentation. Please revise this and put these paragraphs together to deliver a meaningful message. If you are suggesting that these results are similar to what you have showed, you can say that: Our results are consistent with literature. For example, xxx et al. (2019) showed that ….; xxx et al. (2018) showed that ….

Similarly, in line 276-281: you cited two papers but did not say why these are cited? Are you trying to say that these results support your conclusion? If so, you need to say something like: these results are consistent with the trend we found (Fig. xxx).

We agree with you, we have said something like: our results are consistent with literature. For example, xxx et al. showed that ….; xxx et al. showed that …, these results are consistent with the trend we found.

The discussions from line 706 are very confusing. It is difficult to find the logic here. A majority of this is irrelevant so should be removed or distributed elsewhere. For example, for those discussion on PM2.5 – this should go to the part on PM2.5 in “change over time”; Similarly, discussions on SO2 etc. goes to separate sections above.

We agree with you, we have removed majority from line 706.

Remove all of these irrelevant discussions on “clean air acts” in US and UK.

We agree with you, we have removed all of these irrelevant discussions on “clean air acts” in US and UK.

Line 716-724: delete; or you can have a look at Zhang et al. (2019) paper to see and Vu et al. (2019, Atmos Chem Phys) paper to see the effects of meteorology. In the longer term, meteorology is not so important, and the emission change is the dominant contributor to air quality changes. This discussion could be useful

We agree with you, we have deleted Line 716-724. Thank you for your recommendation and help. Papers of Zhang et al. (2019), Vu et al. (2019, Atmos Chem Phys) had a great help for us.

5. Is the manuscript presented in an intelligible fashion and written in standard English?

Reviewer #2: No

We agree with you. We invited Premium Editing of American Journal Experts (AJE) to polish the article. We carefully checked the grammatical errors in the article.

Reviewer #2: This paper is highly improved in this version. The structure of the paper is clear now and more discussion was added in. The data in this paper covers hundreds of cities in China, it provides a whole picture of the evolution of air quality in China in last 6 years. From this standpoint the article is useful and can be used to document levels at present day. Several improvements are needed to be addressed before the manuscript can be considered for publication after minor revision.

1)I would suggest the author give the (possible) reason for the variation of each pollutant. This would help the government to improve the air quality in right direction.

We agree with you. We have given the (possible) reasons for the variations of each pollutant in lines 688-708.

The possible reasons for the changes in PM2.5, SO2, NO2, and CO are related to the strengthening of industrial emission standards. For example, the emission standards of thermal power plants and all emission-intensive industrial sectors (such as steel and cement) have been strengthened. By the end of 2017, more than 95% of China's coal-fired power plants were equipped with flue gas desulfurization (FGD) and selective catalytic reduction (SCR) or selective non-catalytic reduction (SNCR) systems and 71% of the coal-fired power generation capacity reached the "ultra-low emission" standard. In addition, industrial boiler were upgraded and small coal-fired boilers were eliminated, which was important because large-scale operation boilers are widely equipped with SO2 and particulate matter control devices. The elimination of backward industries phases out obsolete or inefficient technology in various industries and allows for structural adjustments. In addition, clean fuel was promoted in the residential sector and advanced stoves and clean coal nationwide were promoted from 2013 to 2016. In 2017, the use of natural gas and electricity to replace coal was further promoted, which affected 6 million households nationwide, of which 4.8 million were located in the Beijing-Tianjin-Hebei area and surrounding areas. The benefits of promoting clean fuels in the residential sector are also obvious throughout the country, and the transportation sector elevated emission standards and imposed mandatory elimination of old vehicles that do not meet emission standards [2].

Such changes would help the government improve the air quality.

2)Line 96-97 on page 5, please rephrase the sentence of “From 2015 to 2019, the national annual mean and annual average median of PM2.5 (unit: μg/m3) were 42.7 and 41, respectively.” I think it should be “The national annual mean and annual average median of PM2.5 (unit: μg/m3) were 42.7 and 41 in 2015 and 2019, respectively.”

I am sorry. We Calculated mean and median. So we said, from 2015 to 2019, the national annual mean and annual average median of PM2.5 (unit: μg/m3) were 42.7 and 41.0, respectively. AJE also checked here again.

3)Line 836 on page 42. The first author should be “B Silver”.

We agree with you, we have changed it. 

Dear Zongbo Shi

Academic Editor

PLOS ONE

Thank you for your help and understanding during these times.

Your reviews's comments were highly insightful and enabled us to greatly improve the quality of our manuscript again.

---

## [Editor Report · Decision Letter 3]

23 Jul 2020

PONE-D-19-35106R3

The spatiotemporal characteristics of the air pollutants in China from 2015 to 2019

PLOS ONE

Dear Guo Ping,

Thank you for submitting your manuscript to PLOS ONE. After careful consideration, we feel that it has merit but does not fully meet PLOS ONE’s publication criteria as it currently stands. Therefore, we invite you to submit a revised version of the manuscript that addresses the points raised during the review process.

Although English has improved, it remains a major issue. I have essentially re-written some parts that are inappropriate or wrong. Please revise and take particular attention at the proper use of units.

We look forward to receiving your revised manuscript.

Kind regards,

Zongbo Shi

Academic Editor

PLOS ONE

Additional Editor Comments (if provided):

Line 11: “simulated data “ – change to “model simulations”

Line 14: “various” – change to “different”

Line 16: add “This led to a major improvement in air quality”

Line 16 – 17: change to “We use the hourly Air Quality Index (AQI) and mass concentrations of PM10, CO, NO2, O3, and SO2 in 362 cities from 2015 to 2019”

Line 35: “through the simulation methods” changed to “using models”

Line 42: “34% SO2, 28% NOx, 26% PM2.5, and 10% volatile organic compounds” change to “SO2 by 34%, NOx by 28%, PM2.5 by 26%, and volatile organic compounds by 10%”

Line 45: “inversion of” change to “retrieved”

Line 54: “upper ozone monitor (OMI)” should this be “The Ozone Monitoring Instrument (OMI)”?

Line 76: delete “In addition, we also found that NO2 has improved as of 2019”

Line 77: delete “propose that it is meaningful to”

Line 87: delete “scientific”

Line 137: add “concentration” after Annual mean

Throughout the manuscript, please add unit to the reported concentration (but no AQI). For example “From a regional perspective, the Northern (58.2), Xinjiang (52.5), and Central (46.6)” should be changed to “From a regional perspective, the Northern (58.2 µg m-3), Xinjiang (52.5 µg m-3), and Central 144 (46.6 µg m-3)”

Line 156: “From 2015 to 2019, the national annual mean PM10 (unit: μg/m3 156 ) was 78.6.” This is incorrect; this has to be changed to “From 2015 to 2019, the national annual mean PM10 was 78.6 µg m-3.”

Line 160: “with highest annual mean.” Change to “with highest annual mean concentration.”

Line 171-172: “Hebei Province, Henan Province and Shandong Province” change to “Hebei, Henan and Shandong Provinces”

Line 178: “Our results showing that higher annual mean PM2.5 concentrations occurred in” : This does not look right; you said earlier PM2.5 is highest in Northern China; now you said that Xinjiang is highest; I think you mean “Our results showing that higher annual mean PM10 concentrations occurred in Xinjiang”

Line 184-186: “considerably different from Zhang et al. because fewer monitoring points were used in Xinjiang in this study, thus, our research results have certain limitations.” Change to “somewhat different from Zhang et al.. This may be attributed to model uncertainty or the limited number of monitoring stations available in Xinjiang in this study.”

Line 187-199: suggest to delete this paragraph

Line 201: “Sanya (14.6), has an annual mean PM2.5 (unit: μg/m3 )” : do remember to revise this: “Sanya, has an annual mean PM2.5 of 14 μg m-3 )”

Line 204: delete “The major cities”

Line 275: “Our results are consistent with the literature.” You have said this many times.

Delete line 275 – 282: This is about CO2 and it has nothing to do with CO

Line 291: “that” change to “the”

Line 299-303: “The fast developing resource and pollution intensive industries along with the ‘Go West’ movement and weak emission controls [30] contributed to the higher rate of increase in NO2 over the Western region from 2005-2013 than over that over the Southwestern, Northern, Eastern, and Southern regions”

Line 312: “We found that NO2 has the opposite trend in cities in the same region.” Change to “We found that NO2 shows a different trend in cities in the same region.”

Line 316: delete “so neighboring areas may exhibit the opposite trend (for example, Hong Kong and the 318 Pearl River Delta)”

Line 324: “were the regions with the highest annual mean (average median) across the country” change to “had a higher annual mean (median) concentration ”

Line 329-333: delete this paragraph. Add a sentence to the end of the last paragraph: “High O3 concentrations in the city of Lhasa of Tibet may be associated with stronger photochemical reactions, vertical mixing and downward transport of stratospheric air mass”

Line 352: “(Table 2-5), (S1-9 Table), (Fig 1-2)” change to “(Table 2-5, Table S1-9; Fig 1-2)”

Line 356: “Our results are consistent with the literature. For example, Guo et al. reported that the number of cities where PM2.5 meets the air quality guidelines of WHO (annual mean 0-10 μg/m3) increased from 0 in 2015 to 1 in 2017, the number of cities where PM2.5 reached target 1 of the transition period of WHO (annual mean 25-35 μg/m3 360 ) increased from 57 in 2015 to 77 in 2017, and the annual mean of 309 cities decreased in 2017 compared with 2015 [18].” Change to “In comparison with 2017 [18], there are more cities which meets the WHO guidelines and the WHO target 1 transition period (annual mean 25-35 μg/m3)”

Line 361: “In addition,” change to: “our results are comparable with literature. For example, “

Line 372: “main reason” change to “key factor”

Line 376: delete “across the country”

Line 377: “value”: what does this mean? Is it “average change”?

Line 378: “The Ali area (-20%) of the Qinghai-Tibet; Hegang (-33.9%) and 379 Baicheng (-18.8%) in the Northeast region; and Haimen (-14.4%) and Jinhua (-12.7%) 380 in the Eastern region are the cities with a faster decline.” Change to “Ali (-20%) of the Qinghai-Tibet, Hegang (-33.9%) and Baicheng (-18.8%) in the Northeast region, and Haimen (-14.4%) and Jinhua (-12.7%) in the Eastern region are the cities with a faster decline.”

Line 381: delete “Our results are consistent with the literature. For example,”

Line 393: “but also from the low emission sources (such as residential coal combustion)” change to “as well as residential sources”

Line 405: “almost the” change to “second highest”

Line 407: “Our results are consistent with the literature. For example,” change to “This is at least partially due to the natural PM sources, such as dust.”: merge this paragraph with the previous one

Line 418-491: “The above findings relate to the relative change of the annual mean of PM2.5 in 419 the five years from 2015 to 2019.” This sentence makes no sense. What “above findings”?

Line 412: delete “Our results 422 are consistent with the literature. For example,”

Line 442: correct the references to figures / tables

Line 443: change “0-20” to “20”

Line 448: PLEASE STOP USING “Our results are consistent with the literature. For example,”: You did this for 18 times. This is how a discussion works. Please discuss the results similarly as suggested earlier for PM2.5. Using literature to help you to explain the results, not to say that yours are consistent with literature.

Line 459-461: Make a similar change as suggested earlier for PM2.5

Line 465: “almost lowest” change to “lower”

Line 675: “Possible reasons for the variations of each pollutant” change to “Factors contributing to the decrease in air pollutant levels”

Line 676: “The possible reasons for the changes in PM2.5, SO2, NO2, and CO are related to the strengthening of industrial emission standards. For example,” Change to “A number of policy actions contributed to the decrease in PM2.5, SO2, NO2, and CO levels. Firstly,”

Line 695: delete

Line 700: “the whole of China is dominated by PM2.5 fine particles, and the spatial distribution shows a concentrated distribution (concentrated in the Northern region, Xinjiang, the Central region, the Weihe River Basin, etc.)” change to “PM2.5 is the most important air pollutant in most regions in China, particularly in the Northern China, Xinjiang, Central China, and the Weihe River Basin)”

---

## [Author Response · Author response to Decision Letter 3]

27 Jul 2020

Dear Zongbo Shi

Academic Editor

PLOS ONE,

Minor revision required [PONE-D-19-35106R4]

Please find attached a revised version of our manuscript The spatiotemporal characteristics of the air pollutants in China from 2015 to 2019.

Your reviews's comments were highly insightful and enabled us to greatly improve the quality of our manuscript again.

Line 11: “simulated data “ – change to “model simulations”

We agree with you, we have changed it.

Line 14: “various” – change to “different”

We agree with you, we have changed it.

Line 16: add “This led to a major improvement in air quality”

We agree with you, we have added it.

Line 16 – 17: change to “We use the hourly Air Quality Index (AQI) and mass concentrations of PM10, CO, NO2, O3, and SO2 in 362 cities from 2015 to 2019”

We agree with you, we have changed it.

Line 35: “through the simulation methods” changed to “using models”

We agree with you, we have changed it.

Line 42: “34% SO2, 28% NOx, 26% PM2.5, and 10% volatile organic compounds” change to “SO2 by 34%, NOx by 28%, PM2.5 by 26%, and volatile organic compounds by 10%”

We agree with you, we have changed it.

Line 45: “inversion of” change to “retrieved”

We agree with you, we have changed it.

Line 54: “upper ozone monitor (OMI)” should this be “The Ozone Monitoring Instrument (OMI)”?

We agree with you, we have changed it.

Line 76: delete “In addition, we also found that NO2 has improved as of 2019”

We agree with you, we have deleted it.

Line 77: delete “propose that it is meaningful to”

We agree with you, we have deleted it.

Line 87: delete “scientific”

We agree with you, we have deleted it.

Line 137: add “concentration” after Annual mean

We agree with you, we have added it.

Throughout the manuscript, please add unit to the reported concentration (but no AQI). For example “From a regional perspective, the Northern (58.2), Xinjiang (52.5), and Central (46.6)” should be changed to “From a regional perspective, the Northern (58.2 µg m-3), Xinjiang (52.5 µg m-3), and Central 144 (46.6 µg m-3)”

Line 156: “From 2015 to 2019, the national annual mean PM10 (unit: μg/m3 156 ) was 78.6.” This is incorrect; this has to be changed to “From 2015 to 2019, the national annual mean PM10 was 78.6 µg m-3.”

We agree with you, we have added unit.

Line 160: “with highest annual mean.” Change to “with highest annual mean concentration.”

We agree with you, we have changed it.

Line 171-172: “Hebei Province, Henan Province and Shandong Province” change to “Hebei, Henan and Shandong Provinces”

We agree with you, we have changed it.

Line 178: “Our results showing that higher annual mean PM2.5 concentrations occurred in” : This does not look right; you said earlier PM2.5 is highest in Northern China; now you said that Xinjiang is highest; I think you mean “Our results showing that higher annual mean PM10 concentrations occurred in Xinjiang”

We agree with you, we have changed it.

Line 184-186: “considerably different from Zhang et al. because fewer monitoring points were used in Xinjiang in this study, thus, our research results have certain limitations.” Change to “somewhat different from Zhang et al.. This may be attributed to model uncertainty or the limited number of monitoring stations available in Xinjiang in this study.”

We agree with you, we have changed it.

Line 187-199: suggest to delete this paragraph

We agree with you, we have deleted this paragraph.

Line 201: “Sanya (14.6), has an annual mean PM2.5 (unit: μg/m3 )” : do remember to revise this: “Sanya, has an annual mean PM2.5 of 14 μg m-3 )”

Yes, we have revised this.

Line 204: delete “The major cities”

We agree with you, we have deleted it.

Line 275: “Our results are consistent with the literature.” You have said this many times. Delete line 275 – 282: This is about CO2 and it has nothing to do with CO

We agree with you, we have deleted it.

Line 291: “that” change to “the”

We agree with you, we have changed it.

Line 299-303: “The fast developing resource and pollution intensive industries along with the ‘Go West’ movement and weak emission controls [30] contributed to the higher rate of increase in NO2 over the Western region from 2005-2013 than over that over the Southwestern, Northern, Eastern, and Southern regions”

We agree with you, we have changed it.

Line 312: “We found that NO2 has the opposite trend in cities in the same region.” Change to “We found that NO2 shows a different trend in cities in the same region.”

We agree with you, we have changed it.

Line 316: delete “so neighboring areas may exhibit the opposite trend (for example, Hong Kong and the 318 Pearl River Delta)”

We agree with you, we have deleted it.

Line 324: “were the regions with the highest annual mean (average median) across the country” change to “had a higher annual mean (median) concentration ”

We agree with you, we have changed it.

Line 329-333: delete this paragraph. Add a sentence to the end of the last paragraph: “High O3 concentrations in the city of Lhasa of Tibet may be associated with stronger photochemical reactions, vertical mixing and downward transport of stratospheric air mass”

We agree with you, we have deleted it and added the sentence to the end of the last paragraph.

Line 352: “(Table 2-5), (S1-9 Table), (Fig 1-2)” change to “(Table 2-5, Table S1-9; Fig 1-2)”

We agree with you, we have changed them.

Line 356: “Our results are consistent with the literature. For example, Guo et al. reported that the number of cities where PM2.5 meets the air quality guidelines of WHO (annual mean 0-10 μg/m3) increased from 0 in 2015 to 1 in 2017, the number of cities where PM2.5 reached target 1 of the transition period of WHO (annual mean 25-35 μg/m3 360 ) increased from 57 in 2015 to 77 in 2017, and the annual mean of 309 cities decreased in 2017 compared with 2015 [18].” Change to “In comparison with 2017 [18], there are more cities which meets the WHO guidelines and the WHO target 1 transition period (annual mean 25-35 μg/m3)”

We agree with you, we have changed it.

Line 361: “In addition,” change to: “our results are comparable with literature. For example, “

We agree with you, we have changed it.

Line 372: “main reason” change to “key factor”

We agree with you, we have changed it.

Line 376: delete “across the country”

We agree with you, we have deleted it.

Line 377: “value”: what does this mean? Is it “average change”?

Yes, it is “average change”, we have changed it.

Line 378: “The Ali area (-20%) of the Qinghai-Tibet; Hegang (-33.9%) and 379 Baicheng (-18.8%) in the Northeast region; and Haimen (-14.4%) and Jinhua (-12.7%) 380 in the Eastern region are the cities with a faster decline.” Change to “Ali (-20%) of the Qinghai-Tibet, Hegang (-33.9%) and Baicheng (-18.8%) in the Northeast region, and Haimen (-14.4%) and Jinhua (-12.7%) in the Eastern region are the cities with a faster decline.”

We agree with you, we have changed it.

Line 381: delete “Our results are consistent with the literature. For example,”

We agree with you, we have deleted it.

Line 393: “but also from the low emission sources (such as residential coal combustion)” change to “as well as residential sources”

We agree with you, we have changed it.

Line 405: “almost the” change to “second highest”

We agree with you, we have changed it.

Line 407: “Our results are consistent with the literature. For example,” change to “This is at least partially due to the natural PM sources, such as dust.”: merge this paragraph with the previous one

We agree with you, we have changed it and merged this paragraph with the previous one.

Line 418-491: “The above findings relate to the relative change of the annual mean of PM2.5 in 419 the five years from 2015 to 2019.” This sentence makes no sense. What “above findings”?

We agree with you, we have deleted them.

Line 412: delete “Our results 422 are consistent with the literature. For example,”

We agree with you, we have deleted it.

Line 442: correct the references to figures / tables

We agree with you, we have corrected it.

Line 443: change “0-20” to “20”

We agree with you, we have changed it.

Line 448: PLEASE STOP USING “Our results are consistent with the literature. For example,”: You did this for 18 times. This is how a discussion works. Please discuss the results similarly as suggested earlier for PM2.5. Using literature to help you to explain the results, not to say that yours are consistent with literature.

We agree with you, we have stoped using this sentence.

Line 459-461: Make a similar change as suggested earlier for PM2.5

We agree with you, we have changed it.

Line 465: “almost lowest” change to “lower”

We agree with you, we have changed it.

Line 675: “Possible reasons for the variations of each pollutant” change to “Factors contributing to the decrease in air pollutant levels”

We agree with you, we have changed it.

Line 676: “The possible reasons for the changes in PM2.5, SO2, NO2, and CO are related to the strengthening of industrial emission standards. For example,” Change to “A number of policy actions contributed to the decrease in PM2.5, SO2, NO2, and CO levels. Firstly,”

We agree with you, we have changed it.

Line 695: delete

We agree with you, we have deleted it.

Line 700: “the whole of China is dominated by PM2.5 fine particles, and the spatial distribution shows a concentrated distribution (concentrated in the Northern region, Xinjiang, the Central region, the Weihe River Basin, etc.)” change to “PM2.5 is the most important air pollutant in most regions in China, particularly in the Northern China, Xinjiang, Central China, and the Weihe River Basin)”

We agree with you, we have changed it.

Dear Zongbo Shi

Academic Editor

PLOS ONE

Thank you for your help and understanding during these times.

Your reviews's comments were highly insightful and enabled us to greatly improve the quality of our manuscript again.

---

## [Editor Report · Decision Letter 4]

6 Aug 2020

The spatiotemporal characteristics of the air pollutants in China from 2015 to 2019

PONE-D-19-35106R4

Dear Guo Peng

We’re pleased to inform you that your manuscript has been judged scientifically suitable for publication and will be formally accepted for publication once it meets all outstanding technical requirements.

Kind regards,

Zongbo Shi

Academic Editor

PLOS ONE

Additional Editor Comments (optional):

There are  a few English issues including inappropriate use of words such as "Quantity" in one of the table captions may be better to be "number", and punctuation issues such as one sentence contains two separate sentences (so a full stop is needed). 
---

## [Editor Report · Acceptance letter]

10 Aug 2020

PONE-D-19-35106R4 

The spatiotemporal characteristics of the air pollutants in China from 2015 to 2019 

Dear Dr. Guo:

I'm pleased to inform you that your manuscript has been deemed suitable for publication in PLOS ONE. Congratulations! Your manuscript is now with our production department. 

Kind regards, 

on behalf of

Dr. Zongbo Shi 

Academic Editor

PLOS ONE